# Towards a cybersecure and privacy enhanced smart grid: A blockchain enabled federated learning framework

Fatima Tariq[1,2], Fatima Anjum[1]*, Xiaochun Cheng[3]*, Shazia Javed[4], Khursheed Aurangzeb[5], Nadia Kanwal[6]

1 Department of Computer Science, Lahore College for Women University, Lahore, Punjab, Pakistan, 2 School of Systems and Technology, University of Management and Technology, Lahore, Punjab, Pakistan, 3 Computer Science Department, Bay Campus, Fabian Way, Swansea University, Swansea, SA1EN, Wales, United Kingdom, 4 Department of Mathematics, Lahore College for Women University, Lahore, Punjab, Pakistan, 5 Department of Computer Engineering, College of Computer and Information Sciences, King Saud University, Riyadh, Saudi Arabia, 6 School of Computer Science and Mathematics, Keele University, Keele, Newcastle, United Kingdom

* fatima.anjum@lcwu.edu.pk (FA); xiaochun.cheng@swansea.ac.uk (XC)

## Abstract

In smart grids, data collection is carried out through smart meters and devices of the Internet of Things, which are installed in the home, allowing to predict the demand for electricity and optimize the distribution of energy. Although the smart grids improve efficiency of operations for end users, they simultaneously present pronounced challenges regarding user privacy and security at the system level. In the context of conventional centralized machine learning, paradigms risk breaching the raw data of consumers, while decentralized paradigms often lack strong mechanisms for verifying identity or ensuring traceability. Existing federated learning systems often lack client level differential privacy, secure aggregation, and decentralized identity protection, leaving them vulnerable to privacy leakage and inference attacks. Blockchain based solutions typically expose model updates or use single layer identifiers. This paper introduces a secure and privacy preserving architecture that combines a dual layer blockchain architecture, federated learning (FL) and central differential privacy (DP) to thoroughly solve these challenges. The proposed system includes a dual layer blockchain system that ensures secure and tamper resistant logging of client interactions and protects client identities by storing salted cryptographic hashes. This design provides both traceability and anonymity, and thus maintains the integrity of participation while obfuscating sensitive identifiers. Privacy is guaranteed by storing raw data in client devices and sending only model updates for central aggregation. At the server side, Gaussian noise is added to the aggregated model parameters to achieve central DP, so as to reduce the risks of inference attacks on user data. Implementation of the proposed framework was performed based on Flower to test the PRECON (Pakistan Residential Electricity CONsumption) dataset, which consists of real-world household electricity consumption data. Multiple machine learning

**Data availability statement:** The data underlying the results presented in the study are available from (https://web.lums.edu.pk/~eig/precon.html).

**Funding:** The authors have been funded by UKRI EPSRC Grant EP/W020408/1 Project SPRITE+ 2: The Security, Privacy, Identity and Trust Engagement Network plus (phase 2) for this study. The authors also have been funded by PhD project RS718 on Explainable AI through UKRI EPSRC Grant funded Doctoral Training Centre at Swansea University. The funders supported the research by reviewing the manuscript and offering technical and editorial suggestions that helped strengthen the presentation of the work. This Research is funded by Ongoing Research Funding Program (ORF-2026-947), King Saud University, Riyadh, Saudi Arabia.

**Competing interests:** The authors have declared that no competing interests exist.

models were benchmarked and out of all the models, Random Forest performed best with the performance metrics of Mean Absolute Error (MAE) of 0.153, Mean Absolute Percentage Error (MAPE) of 0.085 and Mean Squared Error (MSE) of 0.143. The results showed that the proposed framework improved data privacy, preserved the forecasting accuracy and security in smart grid environments.

## Introduction

In the past couple of years, the AI industry has flourished so much that we have shifted from mechanical devices to smart machines. Grids are adapted into Smart grids with smart meters and edge devices to control and manage resources in smart cities. Smart grid (SG) [1] is a radical concept introduced to collaborate power systems' production, distribution, and generation under one umbrella. The aim of SG is to efficiently manage the delivery of energy to the end-users from different generation sources while making use of advanced Artificial Intelligence (AI) algorithms for monitoring of power consumption as well as generation in real time [2,3]. Real time data acquisition in smart grids (SG) is carried out by smart meters, sensors and Internet of Things (IoT) devices, and therefore enables accurate load forecasting and demand management along with energy distribution optimization [4,5]. The systems increase operational efficiency, simplify integrating renewable energy resources, and allow implementing the strategy of responsive demands in a dynamic way. However, the benefits that come along with them raise significant risks associated with data confidentiality, consumer privacy, and the general security of grid infrastructure [6,7]. Smart grids require real-time data collection to be performed through the use of smart meters, sensors and IoT devices to facilitate the proper load forecasting, efficient demand management, and optimization of energy distribution. Whereas these systems enhance the efficiency of operations and allow renewable integration and dynamic demand response, they also raise serious issues of data privacy, consumer privacy, and security of grid infrastructure.

In SG architectures, raw household data are aggregated and sent to centralized servers to be used to train machine learning (ML) models for load forecasting, renewable energy integration and consumption analysis. While centralization is a way to improve the performance of models and to enable the analysis of large data sets, it simultaneously exposes the sensitive data of consumers to potential misuse and privacy violations, data breaches, non-compliance, and cyber attacks [8,9]. Furthermore, the absence of transparent mechanisms for verifying the provenance of data and ensuring input integrity makes these systems susceptible to adversarial manipulation and tampering. In SG environments, the constant exchange of information between smart meters, edge devices, and central servers' results in the exposure of confidential household information, including device usage patterns and meter IDs, to potential tampering, spoofing or unauthorized access [10,11]. Existing communication and logging mechanisms are not sufficient to ensure integrity, hinder manipulation or validate the authenticity of the participating entities.

So, to address this issue, machine learning's approach Federated Learning [12] is a viable solution which consists of edge-enabled AI technologies. FL is a distributed ML technique that allows multiple systems to collaboratively train a global model without sharing, with a central server, any type of data. In the context of SG, FL can be used to improve energy management, load forecasting, and renewable energy forecasting while maintaining the privacy as well as security of data and reducing communication overhead. The application of blockchain technology would also help to increase the security, transparency and auditability of the consumer's sensitive data. The proposed Blockchain enabled Federated Learning framework (BeFL) ensures data protection standards suggested by General Data Protection Regulation (GDPR) [13] and California Consumer Privacy Act (CCPA) [14]. The proposed framework BeFL ensures that raw data remains on client devices, fulfilling data minimization requirements. While unique identifiers are pseudonymized using UUID and SHA-256 with salt. These encrypted hashes are stored on an Ethereum blockchain via smart contracts, enabling immutable audit trails and accountability. The framework avoids direct transmission of personal identifiers and supports opt-in mechanisms. Thus, maintaining compliance with consent, transparency, and de-identification provisions which are mandated by GDPR and CCPA.

In this paper, we propose an integrated framework which combines FL, centralized DP, and a dual layer blockchain architecture to secure users' identities and guarantee that users can participate in collaborative learning. This paper presents a novel privacy and security aware framework that consists of federated learning with blockchain based aggregation for smart grid forecasting. The proposed system maintains data hiding by using a dual layer blockchain to log cryptographic hashes of client sensitive data for data security and applies gaussian noise during model aggregation to meet differential privacy standards [15,16]. By addressing the limitations of prior work, this framework provides a secure and privacy preserving solution for collaborative learning in smart grid environments.

The proposed framework is not restricted to SGs for forecasting, but it can be used in other sectors, where data privacy and trust are of the most importance. In the medical field it can enable model training across hospitals while adhering to data protection laws like Health Insurance Portability and Accountability Act (HIPAA) [17] and GDPR [13]. In smart cities, it can be used for privacy aware mobility or environmental forecasting using edge IoT devices. In the financial sector, the framework can be applied to identify fraud or to optimize the prediction of risk without exposing client sensitive transactional data. Additionally, for the industrial IoT environments, it helps support predictive maintenance and process optimization across the distributed manufacturing units while protecting proprietary information. By maintaining data locality, input origin verifiability, and individual contribution protection, the framework provides a scalable, secure, and privacy-conscious decentralized AI solution for multi domain use.

## Problem statement

In traditional smart grid architectures, raw data is normally obtained from different households which is then aggregated for application of various ML models [9] for forecasting such as load management and renewable energy forecasting. Though this approach of storing data in a central server is effective in terms of model performance. But at the same time these approaches risk the exposure of highly sensitive consumer data to potential cyberattacks, non-compliance or misuse [7,8]. Also, there are high risks of tampering and adversarial manipulation [10,11] due to the unavailability of any type of transparent mechanism for ensuring input integrity and verification of data origin. Overall, this centralized paradigm creates several vulnerabilities in terms of both security and privacy, some of which are also provided below:

- The transmission and storage of raw user data increase the risk of data breach and surveillance.
- Lack of data integrity checks, which makes systems vulnerable to tampering or injection attacks.
- Lack of decentralized validation of the origin of data or identity of participants, which increases lack of trust among participants collaborating to learn.
- Privacy of the individual is compromised during model training.

In order to counter such issues, recent research works have explored FL and blockchain as an alternative. Blockchain ensures the security of sensitive data while the concept of FL introduces a decentralized approach where clients train ML models locally with their own data and then forward only the model updates or parameters to the central server [18,19]. Through this approach, the exposure of raw data is significantly reduced and also the user privacy is maintained as the local data remains on edge devices. However, FL as a standalone method does not completely provide protection against privacy leakage as the shared model parameters may still be breached in order to obtain partial insights into the underlying data distribution [20,21]. In order to counter this, the technique of DP has been utilized by different research works. Through the use of DP, random noise is introduced into the model aggregation which overall limits the amount of information that could be inferred from individual model parameters [15,22]. As such, many existing frameworks apply DP in isolation, which can be crucial for maintaining trust and auditability in collaborative learning environments. However, conventional DP often introduces excessive Gaussian noise to ensure strong privacy, which compromise the accuracy of the training model especially when applied to small datasets. Overall, there is an urgent need of comprehensive framework with enhanced privacy and security measures to not only guarantee user data confidentiality but also to guarantee traceable and tamper resistant provenance of data and verify involvement in collaborative model training.

## Motivation

While scrutinizing existing works, it has been explored that the use of federated learning and differential privacy for smart grid forecasting, falls short in addressing data integrity, input traceability, and consumer privacy preservation.

Some recent studies [5–7,9,10] have attempted to integrate blockchain technology into smart grids, but these are largely focused on transaction logging or access control, not on verifying the legitimacy of data used in machine learning workflows. To date, no existing work effectively combines blockchain-based feature registration, federated local model training, and differentially private aggregation into a single framework designed for privacy, security, and scalability.

This research addresses these gaps by proposing a dual layered framework that:

- Logs hashed feature identifiers on blockchain before training
- Applies federated learning to keep raw data local
- Implements central differential privacy at the server to protect model updates.

This integrated approach ensures that the datasets used in a collaborative learning approach are verifiable, confidential and secure so that they make a strong candidate for implementation in real-world sensitive energy systems.

## Key contributions

In light of the aforementioned challenges, a privacy preserving framework is proposed that helps secure the personal information of power consumers. The proposed framework employs the use of FL, blockchain and central DP. The salient features of the proposed framework are discussed below:

- A dual layer blockchain structure is introduced, in which sensitive identifiers such as client ID (CID)and house numbers (HN) are anonymized using salted cryptographic hash functions before being submitted to smart contracts. This design means that no raw identifiers are ever exposed to the on-chain, significantly reducing the risk of re-identification.
- The framework also integrates duplicate detection under smart contracts to prevent redundant data submissions and increase the efficiency of the blockchain.
- This framework also considers the inclusion of FL to ensure privacy as consumer data is trained on federated clients only. The local parameters from this training are forwarded to a central server for model aggregation while keeping the raw data at client devices.

- To reinforce privacy even more, at the FL server side (on the model aggregation stage) a central differential privacy mechanism is employed through which Gaussian noise is added to the aggregated model parameters to prevent side channel information leakage but preserve the utility of the model.

Furthermore, a thorough analysis is carried out by taking into account some measurements such as gas usage, transaction cost, latency, hash collision rate, raw data leakage, and salt entropy. The results show that the proposed framework achieves robust privacy guarantees with low computational and transactional overhead and is therefore applicable to secure smart grid and IoT applications.

## Proposed work

The proposed framework aims to enhance the security and privacy of smart grid data, without compromising the forecasting accuracy or system performance. The key objectives of this research are as follows:

- To ensure the security and privacy of user collected data from smart grids by integrating blockchain based privacy preservation framework.
- To implement the federated learning approach together with different forms of machine learning models to conduct the training and evaluation across the distributed data sources.
- To integrate a customised FedAvg algorithm thereby increasing the privacy, while transmitting and aggregating models at the central server.

Fig 1 shows a simplified visual representation of the different steps present in our proposed framework. The proposed framework is the novel orchestration of dual layer blockchain, central differential privacy, and federated learning in a smart grid context. The key advantage is the delivery of a framework that is secure, privacy preserving, auditable, and accurate, offering both theoretical guarantees and practical deployability for next-generation energy management systems.

This paper is organized into the following sections. A comprehensive review of related work in the domains of privacy and security preservation in smart grids using federated learning, differential privacy, and blockchain applications is provided in Sect II. While Sect III details the system methodology and presents the proposed framework, including its layered

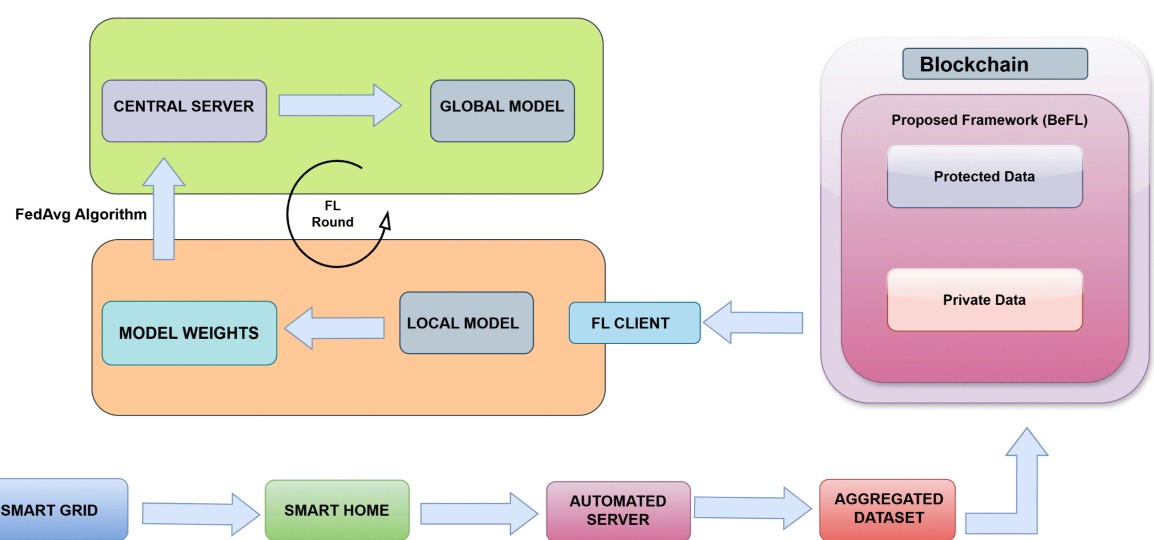

**Fig 1**. Blockchain and FL enabled proposed framework.

design security and privacy preserving mechanisms. Sect IV outlines the experimental setup, including dataset specifications, model configurations, and implementation details. This section also presents the results and performance analysis of the proposed framework. Finally in Sects V and VI conclude the paper and discuss potential directions for future research.

Also provided below in Table 1 is the list of acronyms used and their descriptions, while Table 2 provides the scientific notations and their meanings in this research work.

## Literature work

With the increasing deployment of smart grid infrastructure, securing energy data and ensuring privacy have emerged as vital concerns. This section discusses recent research in three major domains:

- privacy and security preservation in smart grids,
- machine learning and federated learning-based approaches for privacy and
- blockchain applications integrated with FL in the energy sector.

### Privacy and security preservation in smart grids

The distribution of data within the SG poses great challenges to the processing of large volumes of information. Centralized aggregation of data would place huge burdens on communication bandwidth and computational resources [8].

**Table 1**. List of acronyms.

| Acronyms | Description |
| --- | --- |
| AI | Artificial Intelligence |
| BeFL | Blockchain enabled Federated Learning framework |
| B-FRL | Blockchain-Federated Reinforcement Learning |
| CCPA | California Consumer Privacy Act |
| CID | Client ID |
| DP | Differential Privacy |
| FL | Federated Learning |
| GAN | Generative Adversarial Network |
| GDPR | General Data Protection Regulation |
| HGB | HistGradient Boosting |
| HIPAA | Health Insurance Portability and Accountability Act |
| HN | House Number |
| IID | Independent and Identically Distributed |
| IoT | Internet of Things |
| KNN | K-Nearest Neighbor |
| LR | Linear Regression |
| MAE | Mean Absolute Error |
| MAPE | Mean Absolute Percentage Error |
| ML | Machine Learning |
| MLP | Multi-Layer Perceptron |
| MSE | Mean Squared Error |
| PAFLM | Privacy-Preserving Asynchronous FL Mechanism |
| PRECON | Pakistan Residential Electricity CONsumption |
| QFEVAL | Quantum Federated Ensembled Variational Adaptive Learning |
| $R^2$ | Coefficient of Determination |
| RF | Random Forest |
| RPC | Remote Procedure Call |
| SG | Smart Grid |
| SVR | Support Vector Regression |
| UUID | Universally Unique Identifier |

**Table 2**. List of scientific notations.

| Notations | Meaning |
|---|---|
| $H$ | Set of Houses |
| $h_k$ | $k^{th}$ element in $H$ |
| $k$ | Single Household |
| $D_k$ | Household Usage Data |
| $m_k$ | Number of rows (reading) in $D_k$ |
| $n$ | Number of columns (features) in $D_k$ |
| $D'_k$ | Dataset after addition of CID and HN |
| $CID_k$ | CID of $k^{th}$ household |
| $HN_k$ | HN of $k^{th}$ household |
| $D_{agg}$ | Aggregated Dataset |
| $K$ | Household Datasets |
| $\bigcup_{k=1}^{K} D'_k$ | Vertical Concatenation of $K$s |
| $M$ | Total number of rows for all $K$s |
| $n+2$ | Total number of features |
| $\mathbb{R}^{M \times (n+2)}$ | Dimensions of the aggregated $D_{agg}$ |
| $CID_i$ | Original value of CID for $i^{th}$ data entry |
| $HN_i$ | Original value of HN for $i^{th}$ data entry |
| $\mathcal{H}(\cdot)$ | Secure Cryptographic Hash Function |
| $\widehat{CID_i}$ | Hashed CID |
| $\widehat{HN_i}$ | Hashed HN |
| $s_k$ | Secure Random Salt |
| $\parallel$ | Concatenation of $s_k$ to CID and HN |
| $\mathcal{CH}_k$ | Salted Hashed CID |
| $\mathcal{HH}_k$ | Salted Hashed HN |
| $S$ | $l_2$-sensitivity bound |
| $\theta_k$ | Local model Weights or parameter |
| $\varepsilon$ | Privacy Budget |
| $\delta$ | Failure Probability |
| $\sigma$ | Noise Level |
| $N$ | Total number of FL clients |
| $\theta^{agg}$ | Global Model |
| $\bar{\theta}^{agg}$ | Noise Injected Global Model |

Conventional strategies include task partitioning, parallel execution, and data sharing as well as the usage of industrial applications. Nevertheless, many methods are based on direct exchange of proprietary data between the participants, which is insecure and compromises privacy [7,9,10]. The SG infrastructure aims to keep data confidential; hence it is essential for data to be secured in the analytics pipeline to avoid being exploited by malicious users. In 2016, Google introduced FL which is a new method for decentralized training. FL has since been used in a range of research fields and applications including mobile technology, healthcare and the internet of things [5,6].

A secure framework of FL and blockchain has been proposed by a group of researchers [11]. In light of cyber threats, as well as privacy issues, this framework expressly avoids transmission of raw local data. Each node, in the process of the training, updates the individual parameters of its model before broadcasting the refined models to a central authority or to fellow nodes within the network. Although the framework enhances stability and fairness in federated digital-twin modeling, it is limited by residual privacy risks, restricted support for out-of-distribution users, and the smoothing constraints of its LSTM architecture. Lyu et al. [18] suggested a federation based deep learning approach that considers local credibility, fairness, and privacy protection. Privacy ensured with layered encryption techniques and private Generative Adversarial Networks (GANs) were used by nodes to assess input and performance. Nodes also collaborated to evaluate contribution impact. Gradients were encrypted during training to maintain confidentiality, leading to the higher model price. However the framework is evaluated in an independent and identically distributed (IID) environment which is limited

by its reliance on an honest but curious threat model with insufficient robustness against poisoning or Byzantine adversaries. Zhai et al. [23] introduced a novel structure integrating link reliability for predicting power grid energy usage. The framework considered timing, client selection, and delay, but neglected privacy concerns during model training. Sharing regional data during FL posed risks from attackers. The framework has limited its applicability to realistic heterogeneous smart grid environments. Though the impact of non-IID data, large-scale deployments and real wireless dynamics on model accuracy remains unexplored in this paper. Taik et al. [12] proposed FL for smarter energy trading where an individual could predict energy demand and devise a trading plan accordingly. Smart meter analysis displays potential as an evolving field of study.

Security vulnerabilities such as poisoning or backdoor attacks are acknowledged but it has not been experimentally addressed and provided no comparison with state of the art FL baselines. Furthermore, the prosumer decision model and prediction horizons remain simplified, limiting the framework's applicability to large, heterogeneous smart grid deployments. In a publication, [19] researchers employed FL as a method to protect people's privacy for uses such as forecasting behind the meter solar photovoltaic power. With increasing cyberattacks and vulnerabilities in smart meters and distributed grid appliances, research on anomaly detection and theft prevention has also gained a lot of popularity.However this study was limited to only one dataset and one FL model, and does not address adversarial, scalability, or real device constraints, limiting its real world applicability. In research, Alshehri et al. [20] suggested a federated deep learning approach for detecting zero-day attacks over electricity meters. Moreover, Bondok et al. [21] showed that decentralized federated learning frameworks can be resistant to data integrity and trojan insertion attacks. The limitations in both studies rely on simplified simulations with ideal communication, limited attack types and non-IID scalability validation, which limits their applicability to real world smart grid deployments.

## Privacy and security preservation approaches using ML/FL

Most studies concentrate on utilizing encryption methods on the users' original data, which is quite challenging and requires significant effort to execute. This problem can be resolved by using FL which has been utilized in recent research to cohesively train and evaluate ML models without compromising the privacy of data stored on secure nodes. The only information that is transmitted to the server is the ML algorithms' specifications, and this approach has the potential to significantly minimize security and computational challenges associated with centralized ML [15,22,24]. In a paper [16] scholars suggested Privacy-Preserving Asynchronous FL Mechanism (PAFLM) for an edge network which could enable numerous edge nodes to perform more effective FL without disclosing their private data. The suggested technique, as compared to conventional distributed learning, compresses communication between nodes and the parameter server during training without sacrificing accuracy. Although the paper aims to reduce communication costs, it does not model real-world IoT network challenges, such as intermittent connectivity, packet loss, bandwidth fluctuation, or device dropout, even though IoT environments are inherently unstable. A group of Chinese academics developed the PEFL protocol [25] , which protects the confidentiality of training data, both during and after the training process and even when numerous entities overlap with one another. The paper states PEFL works for large-scale scenarios, but no experiments are conducted with thousands of clients or high-dimensional industrial gradients. In a paper [26], by combining additively homomorphic encryption with differential privacy, researchers devised a protocol for an effective and privacy preserving federated deep learning system based on the stochastic gradient descent technique. This protocol would be able to stop data from being exploited by untrusted parties. However, the paper does not consider network delays, packet loss, dropout patterns, or scalability bottlenecks in real-world FL communication. Machine learning and FL techniques have become influential in preserving data privacy during smart grid analysis [27]. The DP$^2$-NILM model introduced by Shao et al. [28] utilizes FL for non-intrusive load monitoring without exposing users' data. Similarly, pNILM by Ren et al. [29] integrated deep neural networks with encryption for privacy throughout the learning process. Alghamdi et al. [30] proposed Fed-SAD,

 

a secure aggregation framework to safeguard forecasting models from poisoning attacks. While [28–30] papers offer useful insights into applying federated learning for privacy-preserving energy forecasting, their findings are limited by small controlled datasets, ideal communication assumptions, and minimal adversarial modeling, leaving scalability, real-world robustness, and edge-device feasibility insufficiently validated. Cheng et al. [31] proposed a horizontally distributed FL approach that enables private edge cloud energy disaggregation. Additionally, Wang et al. [32] incorporated GRU and ensemble learning within FL to detect electricity theft, reinforcing the potential of hybrid learning structures. Ren et al. [33] further introduced a quantum enhanced FL framework, Quantum Federated Ensembled Variational Adaptive Learning (QFEVAL), to assess dynamic grid threats while minimizing communication overhead. However, all three frameworks [31–33] rely on controlled datasets, ideal communication conditions, limited adversarial modeling, and no real-device validation. While leaving their scalability, robustness, and practical deployment in large, heterogeneous smart-grid environments insufficiently demonstrated.

## Blockchain incorporation in smart grid privacy and security

Over the past couple of years, several studies have been conducted over the topic of privacy concerns and data security in smart grid. Blockchain is an emerging technology [34,35] which researchers are using for security and authorization. The use of blockchain technologies in combination with FL [11,36–38] is achieving more attention for ensuring integrity, accountability, and privacy [39,40] in smart grid systems. In [41] researchers presented a blockchain based framework for authentication and authorization in IoT as a solution to protect data security as well as privacy in smart cities. Yu et al. [42] proposed a dual robust federated digital twin framework with blockchain for real time grid monitoring. Zhang et al. [43] proposed FedGrid, a model that combines the FL and blockchain mechanism, for ensuring energy usage optimization in security. Goodman et al. [44] underlined the contribution of decentralized computation in the increased edge resilience and data protections. While these studies demonstrate valuable advancements in applying FL, blockchain, and digital-twin technologies to enhance privacy, forecasting accuracy, and coordination in smart-grid systems, their reliance on simplified datasets, idealized network conditions, and limited adversarial modeling leaves scalability and cyber-resilience in real heterogeneous deployments insufficiently validated.

Together, these studies offer foundational insights for the design and development of the proposed PPS framework, combining privacy preserving computation, distributed intelligence, and secure blockchain integration.

Table 3 presents a structured analysis of existing literature, categorized into seven research areas aligned with the architectural layers of the proposed BeFL framework. The studies cover key aspects such as privacy preserving data collection at the edge, federated load forecasting, adversarial threat mitigation, and blockchain-enabled integrity and billing.

Table 3 shows the significant contributions that can be found in the surveyed literature as it relates to federated energy prediction, blockchain enabled integrity and smart metering. Nevertheless, several critical limitations are still pervasive. Foremost, there is a lack of end to end comprehensive privacy enforcement with limited implementation of differential privacy on the client or of secure aggregation protocols for federated updates. Moreover, mechanisms for protecting identity through multi-layer hashing, role based access control, and an adaptive trust based client selection are largely underexplored. A few studies are referring to update level tamper evidence or to enable dynamic participation under an adversarial environment.

In order to mitigate these weaknesses, the proposed BeFL framework adopts a multilayer architecture, which combines federated learning, differential privacy, blockchain based logging and secure smart contract validation. Unlike previous models, it provides update level confidentiality, hash-based identifier separation, and dynamic, trust aware client selection for secure aggregation. Additionally, it has a dual layer blockchain architecture (PrivateLayer and ProtectedLayer) to prevent re-identification, and allows fine grained access control using smart contracts, which alleviates the major privacy, security, and scalability limitations identified in previous studies.

**Table 3.** Critical summary of research areas and identified gaps aligned with the proposed BeFL framework.

| Thematic Area of Research | Core Contributions and Technical Focus | Corresponding Layer in Proposed BeFL Framework | Representative Studies | Limitations / Research Gaps |
|---|---|---|---|---|
| Edge Intelligence & Privacy-Preserving Data Collection | Utilizes on device FL and smart meters (NILM) to enable distributed energy monitoring and privacy preserving computation | Edge Device Layer, Secure Smart Meters, NILM Modules | [1,2,4,8,27] | 1. Limited use of differential privacy. 2. No blockchain-based audit trail for data sharing. 3. Static client participation; lacks trust-based selection |
| Federated Learning for Load Forecasting and Disaggregation | Implements FL for decentralized load forecasting and appliance level disaggregation. | FL Model Training Layer, Forecasting Engine | [1,3,4,9,35] | 1. Global model parameters often exposed. 2. Absence of secure aggregation or DP-FedAvg 3. No tamper evident model update logging |
| Security Against Adversarial Attacks in Federated Systems | Proposes defences against data poisoning, inference and Trojan attacks in FL systems. | Threat Detection Unit, Secure Aggregation Mechanism | [5–7,9] | 1. Focused on detection rather than prevention 2. No blockchain backed verification 3. Lacks identity and access control layers |
| Blockchain Integration for Trust and Integrity | Combines blockchain and FL to record updates, ensure data provenance and decentralized trust. | Blockchain Ledger Layer, Integrity Verification Unit | [10,11,18,39,40,45] | 1. Replicates model/state on-chain → privacy leakage 2. No dual-layer (CID/HN) identity separation 3. Absence of differential privacy mechanisms |
| Smart Contract-Based Billing and Consent Mechanisms | Uses smart contracts for automated billing, energy trading, and user consent. | Blockchain Billing Layer, Consumer Authentication and Access Control | [11,12,18,23,46] | 1. Consent and billing models not privacy-hardened 2. Lack of encrypted identifier mapping 3. No cross-layer audit or DP integration |
| Hybrid Edge–Cloud Coordination & Model Personalization | Explores hybrid FL for edge–cloud orchestration and personalized model updates. | Edge–Cloud Coordinator Layer, Personalized FL Module | [4,10], [23,47,48] | 1. Insufficient handling of heterogeneous clients 2. Missing secure aggregation across layers 3. No dynamic participation control |
| Next-Generation Grid: Quantum FL, Digital Twins, Grid 2.0 Vision | Presents futuristic directions such as quantum FL and federated digital twins for smart grids. | Future Expansion Layer, Privacy-by-Design and Scalable Intelligence | [10,11,23] | 1. Conceptual; lacks implementation evidence 2. No concrete privacy or identity framework 3. No evaluation of resilience under adversarial nodes |

The novelty of this study lies in the design and implementation of a multi-layer blockchain enabled federated learning framework with central differential privacy for smart grid energy forecasting. Unlike existing works that address security or privacy in isolation, our BeFL framework provides a domain specific integration that simultaneously ensures identity protection, secure aggregation, and privacy preserving forecasting. The literature work shows that prior studies typically

1. rely on single layer blockchain with limited confidentiality,
2. apply local DP only at client-side aggregation, or
3. lack comprehensive evaluation of both privacy and forecasting utility together

Our work addresses these gaps by introducing a dual blockchain architecture, central DP for improved accuracy privacy trade-off, and joint evaluation of security and forecasting performance.

## Methodology

A secure and privacy preserving framework (BeFL) for smart grid energy forecasting is presented in this paper, based on the integration of federated learning and a blockchain-backed mechanism for data integrity. The proposed BeFL framework ensures that sensitive user data never leaves the place of origin, while at the same time ensuring a verifiable record for participation and accountability on blockchain. The proposed BeFL framework is structured into three core layers designed to achieve data integrity, privacy protection, and robust modeling. These layers include

- Data pre-processing and aggregation,
- Blockchain integration for privacy preservation, and
- Local model training using FL enhanced with centralized DP Strategy.

Each layer plays a crucial role in the system that ensure clarity, security, and practical effectiveness. Fig 2 shows a visual description of the different layers of the proposed BeFL model. The novelty of the suggested framework lies in the multi-layered structure that combines autonomous data collection, dual-layer hierarchical blockchain, and federated learning with centralized differentiation privacy. In particular, data-collection layer ensures safe pre-processing at the edge, the blockchain layer uses SHA-256 with salting and separates identifiers between the private and the protected layers and the FL layer uses centralized differential privacy only at the aggregation level. Together, these innovations provide a secure, auditable, and privacy compliant solution to SG for load forecasting, which provides better model utility compared to traditional solutions.

### Layer 1: Data pre-processing and aggregation

The process starts in any SG where various smart homes IoT devices capture detailed electricity usage data from different appliances located in different rooms. The data from an individual home is sent through a communication hub to an automated server that structures the data onto an aggregated file. The proposed BeFL framework assumes that there are clusters, consisting of 5-10 houses, in a smart grid. The research considers a single cluster of houses for this proposed framework. This cluster can be represented mathematically ($H$) as shown in Eq 1.

$$H = h_1, \ h_2, \ h_3, \ .... \ h_k \tag{1}$$

From the above Eq 1, $h_k$ represents a house which is generating electricity and the usage data is represented by $D_k$. This data is structured with a real number matrix comprising of $m_k$ observations which are continuous numerical values. This data can be represented mathematically as shown in Eq 2.

$$D_k \in \mathbb{R}^{m_k \times n} \tag{2}$$

The matrix with $m_k$ represents rows where each row being an individual observation and $n$ is number of columns with each column representing a distinct feature. These attributes, or features include date and time, electricity usage consumption in kilowatts (Usage_kW) and electricity consumed in different rooms of the house like bedroom, kitchen and drawing room as well as appliances such as AC, UPS. Together, these features provide a detailed representation of a household's energy profile.

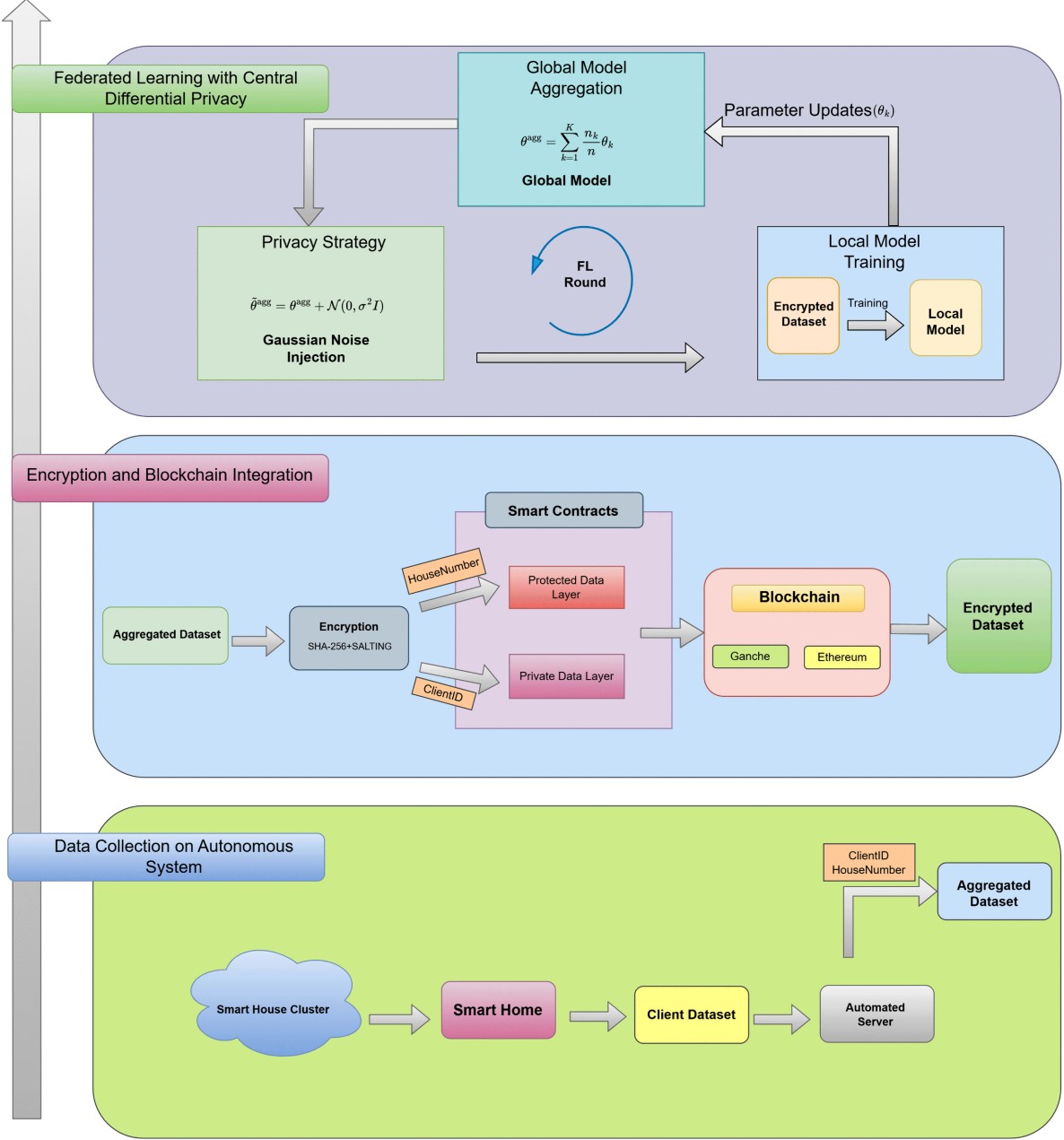

**Fig 2**. **Blockchain enabled FL framework for security and privacy preservation.**

The aggregation process [49], done at the automated server, begins with reading each household's data using a Python script. For traceability and identification, each dataset entry is augmented by generating unique identifiers: a ClientID (CID) and a HouseNumber (HN). These identifiers were created programmatically using UUID (Universally Unique Identifier) generation functions. UUIDs are 128-bit standardized identifiers designed to be globally unique without central coordination. In this framework, UUID version 4, which relies on random or pseudo-random number generation, is

employed to ensure that each identifier is statistically unique. The formal representation of the identifiers is shown in Eq 3:

$$D'_k = \left[D_k \mid CID_k \cdot 1_{m_k}, \; HN_k \cdot 1_{m_k}\right] \tag{3}$$

The transformed dataset from each household is then concatenated to form a single, aggregated dataset. It is ensured that during this process, the original data is not lost, omitted or altered. The aggregated dataset is represented in Eq 4:

$$D_{agg} = \bigcup_{k=1}^{K} D'_k, \quad D_{agg} \in \mathbb{R}^{M \times (n+2)}, \; where \; M = \sum_{k=1}^{K} m_k \tag{4}$$

From the above Eq 4, the notations can be described as follows:

- $D'_k$: This represents the dataset from household $k$ after expansion with two additional columns, CID and HN, resulting in $n + 2$ columns, where $n$ is the total number of features.
- $\bigcup_{k=1}^{K} D'_k$: This refers to the vertical concatenation of all $K$ household datasets.
- $D_{agg}$: This represents the aggregated dataset, consisting of all $D'_k$, having $M$ rows and $n + 2$ columns.
- $\mathbb{R}^{M \times (n+2)}$: This shows that the aggregated dataset is a real-valued matrix. While $M = \sum_{k=1}^{K} m_k$, refers to the total number of rows.

The raw aggregated dataset ($D_{agg}$) is not stored on the blockchain, as doing so would be computationally infeasible and contrary to the privacy principles of federated learning. Instead, only cryptographic hashes of the unique identifiers specifically, CID hash and HN hash, are logged on the chain for data integrity, traceability, and audit purposes. These hashes act as verifiable proofs to ensure that the datasets used in model training are unaltered, without exposing the actual energy consumption data or compromising user privacy. The aggregated file represents the collection of multiple individual datasets, each augmented with unique identifiers, are combined vertically into a single comprehensive aggregated dataset which is suitable for further analysis or modeling. The aggregated dataset is then stored as a separate CSV file for further analysis.

The visual representation of layer 01 where the data is aggregated from different smart homes is displayed in Fig 3. In the proposed BeFL framework, smart meters and IoT-enabled devices feed data into a pre-processing hub that prepares datasets without exposing raw consumer information. Unlike conventional systems where data is centrally stored, our design ensures pre-processing happens at the edge, minimizing risks of raw data leakage.

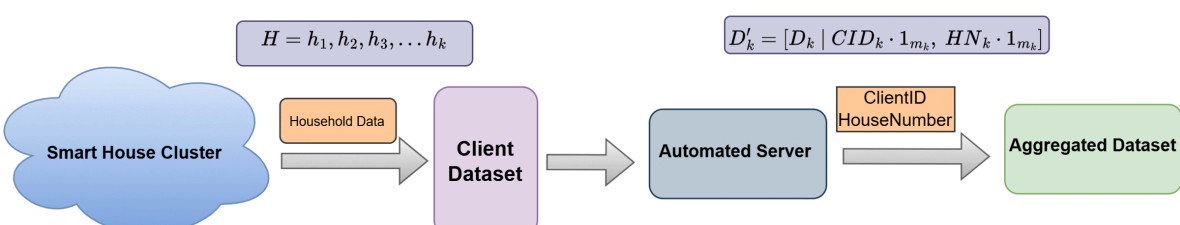

**Fig 3. Layer 1: Data pre-processing and aggregation.**

**Layer 2: Encryption and blockchain integration**

The next layer in the proposed framework consists of two phases. The first phase is the encryption of the secure identifiers that were integrated in the previous layer. The second phase would be the passing of the encrypted identifiers into the Ethereum blockchain. The first phase is initiated with the use of Secure Hash Algorithm (SHA-256) to encrypt the identifiers, CID and HN. The SHA-256 was chosen because of its strong cryptographic properties such as pre-image resistance, determinism and collision resistance. Through the use of SHA-256, each identifier would be converted to a unique and fixed 256-bit length hash [50,51].

The application of the hashing on the identifiers is represented in Eq 5:

$$\widehat{CID}_i = \mathcal{H}\left(CID_i\right), \quad \widehat{HN}_i = \mathcal{H}\left(HN_i\right) \tag{5}$$

- $\mathcal{H}(\cdot)$: Represents a secure cryptographic hash function. For this framework, it specifically refers to SHA-256.
- $CID_i$ and $HN_i$: Denote the original values of the Client ID and House Number for the $i$–$th$ data entry.
- $\widehat{CID}_i$ and $\widehat{HN}_i$: Are the resulting hashed versions of the identifiers.

It would also ensure data security in such a way that if a single input is modified, the resultant hash would be completely different. To further ensure identifier-level privacy, each client generates a cryptographically secure random salt $s_k$, which is concatenated with the client's unique identifiers such as the Client ID ($CID_k$) and House Number ($HN_k$). These concatenated values are then hashed using a secure cryptographic hash function $\mathcal{H}(\cdot)$ (e.g., SHA-256), resulting in the an anonymized representations as shown in Eq 6:

$$\mathcal{CH}_k = \mathcal{H}(s_k \parallel CID_k), \quad \mathcal{HH}_k = \mathcal{H}(s_k \parallel HN_k) \tag{6}$$

The Eq 6 represents the process of generating anonymized identifiers using a salted hashing mechanism. In this formulation, $\mathcal{H}(\cdot)$ denotes a secure cryptographic hash function (e.g., SHA-256), while $s_k$ is a cryptographically generated random salt unique to each client $k$. The operator $\parallel$ signifies the concatenation of the salt and the corresponding identifier string.

By applying this approach, even identical identifiers across different clients will produce distinct hash values due to the uniqueness of the salt. This design enhances privacy by ensuring that the original client identifiers—such as the Client ID ($CID_k$) and House Number ($HN_k$)—cannot be reverse engineered or linked through deterministic hashing. Only the resulting salted hashes ($\mathcal{CH}_k$, $\mathcal{HH}_k$) are submitted to the blockchain, while the raw identifiers and salts remain securely stored off-chain. This ensures strong resistance against linkage attacks and supports compliance with privacy preserving data handling practices.

The resultant hashed identifiers serve as a privacy preserving mechanism because of the following factors:

- Ensure that identifiers are obfuscated before they are stored or transmitted.
- Salt Hashing is irreversible; as it would not be possible to reconstruct the original ID from its hash.
- Even a tiny change in the input (for example, one character) would yield a completely different hash output.

The salt hashing of the identifiers is crucial for the proposed framework, as the user data would remain confidential while maintaining consistent transparency. The resulting salted hash values are submitted to the blockchain smart contracts, without ever exposing raw identifiers on-chain. This design ensures unlinkability, resists brute-force attacks, and maintains compliance with privacy by design principles. Furthermore, each smart contract includes logic to check for duplicate hash entries. If an identical salted hash already exists on-chain, the transaction is skipped to avoid redundant writes and maintain chain integrity.

After the process of hashing is done, the encrypted identifiers are securely transferred and stored onto an Ethereum blockchain environment. This is achieved through the use of Ganache for simulation purposes. The blockchain is supported by two defined smart contracts, executed via Python scripts interfacing through Web3.py. The two defined smart contracts are named "PrivateLayer" and "ProtectedLayer" respectively are then deployed as follows:

- The PrivateLayer smart contract records the association between original Client IDs and hashed House Numbers.
- The ProtectedLayer smart contract maintains records linking original House Numbers with hashed Client IDs.

These smart contracts can also be represented through Eq 7

$$f_{\text{PrivateLayer}} : \text{CID}_i \mapsto \widehat{\text{HN}}_i, \quad f_{\text{ProtectedLayer}} : \text{HN}_i \mapsto \widehat{\text{CID}}_i \tag{7}$$

For each blockchain transaction, careful nonce management ensures that there is conflict free and sequential processing. Different metrics such as transaction hashes, nonce values, gas consumption and transaction duration are systematically recorded to ensure transparency, optimization of blockchain efficiency and facilitation of auditing. Fig 4 shows the working in layer 2 that deals with blockchain integration and implementation where a two-tier blockchain structure is in control of the private layer and securely stores hashed household identifiers, while the protected layer manages anonymized client IDs. This separation ensures that adversaries cannot correlate household data with client identity, thereby mitigating re-identification and linkage attacks while retaining auditability. The dual-layer blockchain ensures end-to-end integrity and confidentiality of smart grid participants. Sensitive identifiers are cryptographically separated across layers, which is a departure from single chain models that expose more attack surfaces.

The dual layer architecture enables a two way authentication of client registration with the maintenance of pseudonymity. The clients are only allowed to participate in training once they have been registered successfully in both layers. In the pre-processing step, the set of associated mappings are uploaded to the corresponding blockchain contracts before any local training can take place. Importantly, raw identifiers are never sent, and hashed values in combination with salted entropy are used, which is compliant with the requirements of GDPR and CCPA privacy regulation. This dual layered architecture is the foundation of non-repudiation, integrity of inputs and a strong client authentication mechanism besides reducing the risk of identity exposure. The system uses privacy preserving decentralization by splitting the hashed mappings of two layers of smart contracts without reducing auditability or trust required in collaborative learning.

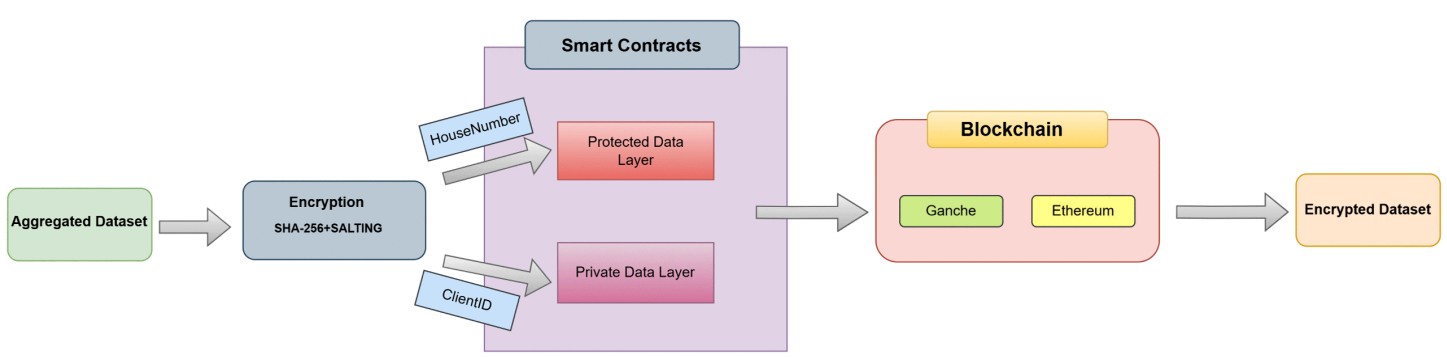

**Fig 4**. **Layer 2: Encryption and blockchain integration.**

## Layer 3: Federated learning with differential privacy strategy

After the encrypted dataset is obtained from the blockchain phase, the dataset is sent to FL clients for furthur processing. Each client, using the encrypted dataset, performs local model training using different ML models such as Random Forest, XGBoost, HistGradientBoosting, MLP, SVR, Ridge Regression and KNN. Before the training begins, some pre-processing steps are applied which includes the division of the dataset into different subset for different FL-clients. After this, the encrypted identifiers, CID and HN, are removed to maintain confidentiality. Other pre-processing measures taken were the feature scaling and statistical imputation for ML models like MLP and SVR, as these models are sensitive to feature magnitude. These pre-precessing steps ensure optimal local performance and consistent data quality. [52].

After pre-processing, each FL client then minimizes the defined loss function $\mathcal{L}(\cdot)$ using its dataset subset $D_k$. This minimization results in the generation of local model weights or parameters $\theta_k$, which is represented in Eq 8:

$$\theta_k = \arg\min_{\theta} \frac{1}{|D_k|} \sum_{(x_i, y_i) \in D_k} \mathcal{L}(f_{\theta}(x_i), y_i) \tag{8}$$

These local parameters or weights are then sent to a central server using FedAvg algorithm as shown in Eq 9:

$$\theta^{\text{agg}} = \sum_{k=1}^{K} \frac{n_k}{n} \theta_k, \quad \text{where } n = \sum_{k=1}^{K} n_k \tag{9}$$

**Custom FedAvg Strategy:** Once the local weights are aggregated by the central servers to achieve a global model, a custom FedAvg strategy is applied on the weights. The defined custom strategy extends the logic of the standard FedAvg algorithm in a two-step process. The first step involves the constraining global sensitivity by clipping the norm of gradient vectors for each client to a fixed bound $S$. The second step is the injection of Gaussian noise to the aggregated parameters (global model). This noise injection is similar to DP, where each clients adds noise to their local parameters before sending to server. However, for this research, centralized DP approach is utilized where the server injects noise into the aggregated parameters.

The formal representation of central DP, where Gaussian noise is injected into the global model is represented in Eq 10:

$$\tilde{\theta}^{\text{agg}} = \theta^{\text{agg}} + \mathcal{N}(0, \sigma^2 I) \tag{10}$$

The $\sigma$ represents the noise level which is adjusted to fulfill the centralized DP $(\varepsilon, \delta)$ through the standard bound as shown in Eq 11:

$$\sigma \geq \frac{2S \cdot \sqrt{2 \ln\left(\frac{1.25}{\delta}\right)}}{N \cdot \varepsilon} \tag{11}$$

Where:

- $S$ represents the $l_2$-**sensitivity bound** which has been set to 1 for this research.
- $\varepsilon$ is the **privacy budget** which has been empirically set to 1.0
- $\delta$ represents the **failure probability** which has been fixed at $10^{-5}$
- While, $N$ is the total number of clients participating

The selection of the above values to ensure a strong balance between model utility and privacy leakage safeguard. After the injection of noise, the final global model ($\tilde{\theta}^{\text{agg}}$) is then transmitted to different FL clients for another FL round.

The above discussed methods ensure that the identifiers of the participating clients is preserved and obscured during collaborative learning. The integration of the custom FedAvg strategy that employs central DP presents a key novel component in the proposed BeFL framework as compared to traditional DP-FedAvg methods.

This strategy ensures that the influence of any single client's data is obscured, while collaborative learning remains effective and privacy preserving. The integration of central DP, norm-bounded aggregation, and post aggregation noise injection constitutes our key enhancement over traditional DP-FedAvg methods. A visual representation of the Cognitive layer is presented in Fig 5.

The step by step implementation of the proposed framework is described in Algorithm 1 that combines the blockchain with the federated learning and central differential privacy. The procedure starts with the local processing of the distributed dataset. During which unique client and household identifiers (Client ID, House Number) are generated and are cryptographically hashed with the assistance of salted SHA-256 to ensure anonymity and integrity. These hashed identifiers are then stored securely on the Ethereum blockchain in two dedicated smart contracts which ensure verifiability and unlinkability. These hashed records are then aggregated and distributed to federated clients, where local training occurs without exposing raw data. From each client individual training, local parameters ($\theta_k$) are calculated and sent to the central server. The central server then aggregates all the client-side parameters into a global model ($\theta^{agg}$). The global model is then introduced to noise that comes from a Gaussian distribution to enforce ($\epsilon, \delta$) central DP. This injection of noise results in a final global model ($\widetilde{\theta^{agg}}$).

This proposed design achieves three objectives simultaneously:

- secure identity management through blockchain,

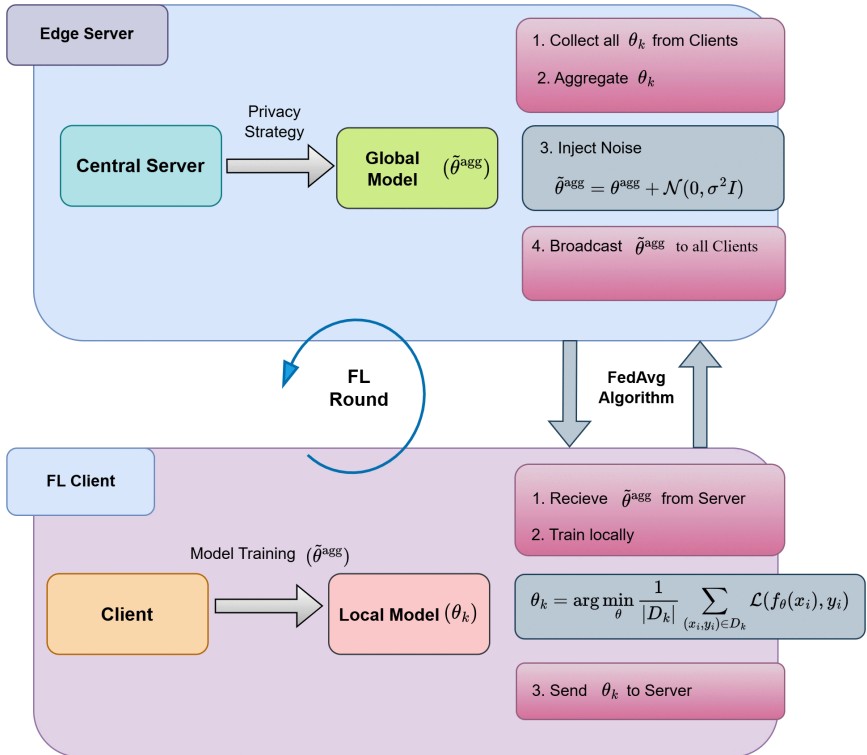

**Fig 5**. Layer 3: Federated learning with differential privacy strategy.

**Algorithm 1 Algorithm: Blockchain enabled federated learning proposed framework.**

**Input:** Distributed datasets $\{D_k\}_{k=1}^{K}$, privacy parameters $\epsilon, \delta$, sensitivity $S$

**Output:** Differentially private global model $\widetilde{\theta^{\mathrm{agg}}}$

```
1:  for each client k = 1, 2, …, K  do
2:      Read raw dataset D_k
3:      Generate UUID-based identifiers CID_k, HN_k
4:      Augment dataset: D_k → D'_k with CID_k, HN_k
5:      Generate random salt s_k, compute salted hashes:
6:      CH_k = H(s_k ‖ CID_k),   HH_k = H(s_k ‖ HN_k)
7:      Submit CH_k, HH_k to blockchain smart contracts
8:  end for
9:  Aggregate all records: D_agg = ⋃_{k=1}^{K} D'_k
10: Distribute D'_k subsets to client-side federated clients
11: for each federated learning round do
12:     for each client k do
13:         Train local model and compute parameters θ_k
14:     end for
15:     Server aggregates models: θ^agg = ∑_{k=1}^{K} (n_k/n) θ_k
16:     Add Gaussian noise: θ̃^agg = θ^agg + N(0, σ²I)
17: end for
        return θ̃^agg
```

- decentralized training via federated learning, and
- strong privacy guarantees through central differential privacy.

In Fig 6 the workflow diagram of proposed BeFL framework which comprises of three main phases: data collection and pre-processing, blockchain based secure storage, and federated learning with central differential privacy is illustrated. The main contribution of this work is a layered framework for smart grid load forecasting that combines autonomous data collection, a dual layer hierarchical blockchain, and federated learning with central differential privacy. By introducing a dual blockchain structure (private and protected layers with SHA-256 + salting) and applying differential privacy only at the aggregation stage, the framework ensures confidentiality, integrity, and high model utility. This design provides a secure, auditable, and practically deployable solution that outperforms existing single layer blockchain and client-side DP-FL approaches.

## Performance evaluation and analysis

In this section, a comprehensive evaluation of the proposed framework is provided in the perspective of blockchain efficiency, and model performance based on different evaluation metrics.

In regards to the proposed hierarchical layered blockchain layer, the efficiency is determined through aspects such as average transaction durations and gas usage. While, for the evaluation of model performance, different ML models were utilized and then compared in both a FL based scenario and a non-FL based scenario to test the effectiveness of the proposed framework. Also, a comparative analysis was also conducted with the proposed framework against recent published articles.

### Dataset

For the experimentation of the proposed framework, this research focused specifically on energy consumption dataset present in the context of developing countries. After a comprehensive review, the PRECON (Pakistan Residential Electricity CONsumption) dataset [53] was selected and utilized. This dataset consists of consumption records of high energy

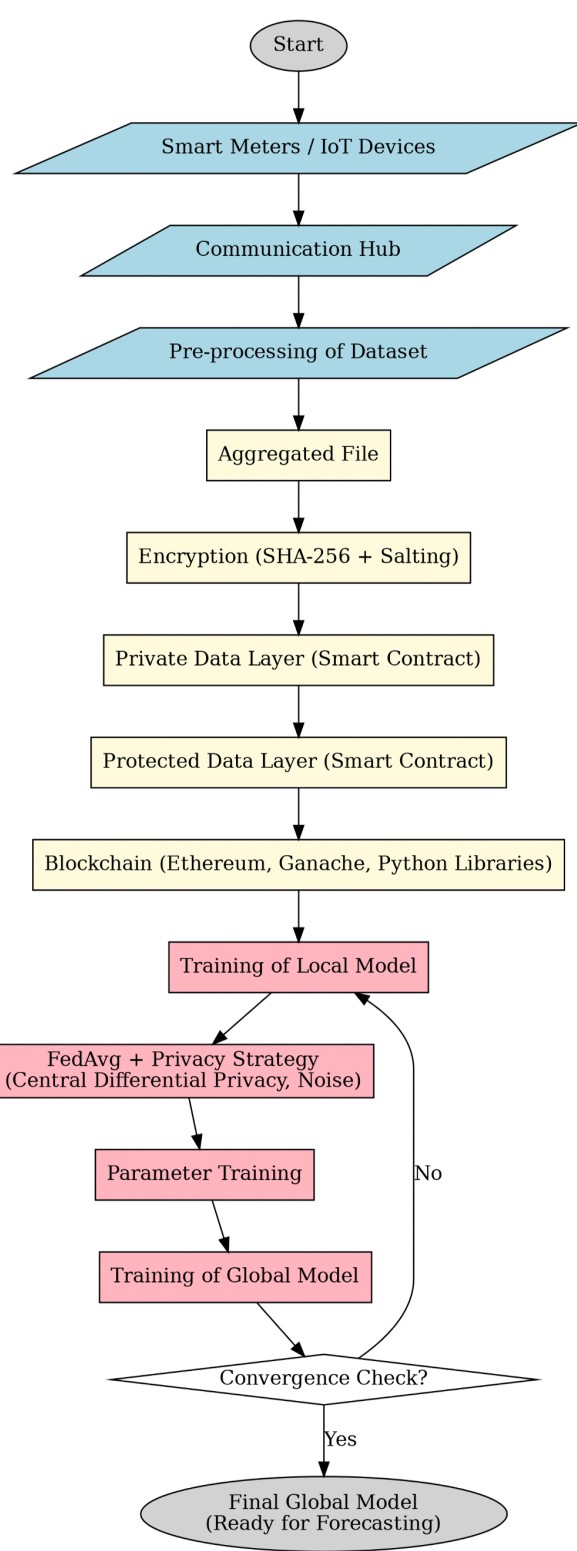

**Fig 6. Workflow diagram of proposed BeFL framework.**

use appliances from different households over a 1-year period. The suggested methodology is based on the application of a multi layer blockchain enabled federated learning architecture with central differential privacy to the dataset called PRECON in order to demonstrate that robust security and privacy protection mechanisms can be achieved without sacrificing the accuracy of the forecasts. The validity of these results is supported by the diversity of the samples of homes in PRECON, which includes 42 households with different demographics, appliances and wiring configurations and makes them representative of real world smart grid environments. The generalizability stems from the fact that consumption patterns, appliance usage, and demographic factors captured in 2018 remain structurally consistent in 2025, making the findings applicable to other South Asian and developing country contexts. Although collected in 2018, PRECON remains the most comprehensive, fine grained and publicly available dataset from the region, widely recognized in literature, which rationalizes its continued use and ensures reproducibility with international benchmarks.

Fig 7, shows a heatmap for a subset of a single household data, obtained from the PRECON dataset. The Y-axis values represents the timestamp when the reading was taken while X-axis represents the different features or places such as the AC in the drawing room (AC_DR_kW), UPS, lounge room (LR_kW) and kitchen. The Usage_kW represents the total consumption of all the features combined at each timestamp. The color gradients reflect the magnitude of each reading, where a darker shade represents higher power usage.

## Experimental setup

For the experimental setup, different tools and libraries were utilized for each of the proposed layers which are discussed below.

**Data pre-processing and aggregation.** The first layer of the proposed framework consisted of obtaining the PRECON dataset which initially involved the inclusion of 42 (.csv) files with each file corresponding to a single household data. After obtaining the dataset, clusters were made with each cluster involving 4-5 household (.csv) files. The next step after this was the generation of unique identifiers in the form of ClientID and HouseNumbers using UUID-based randomization

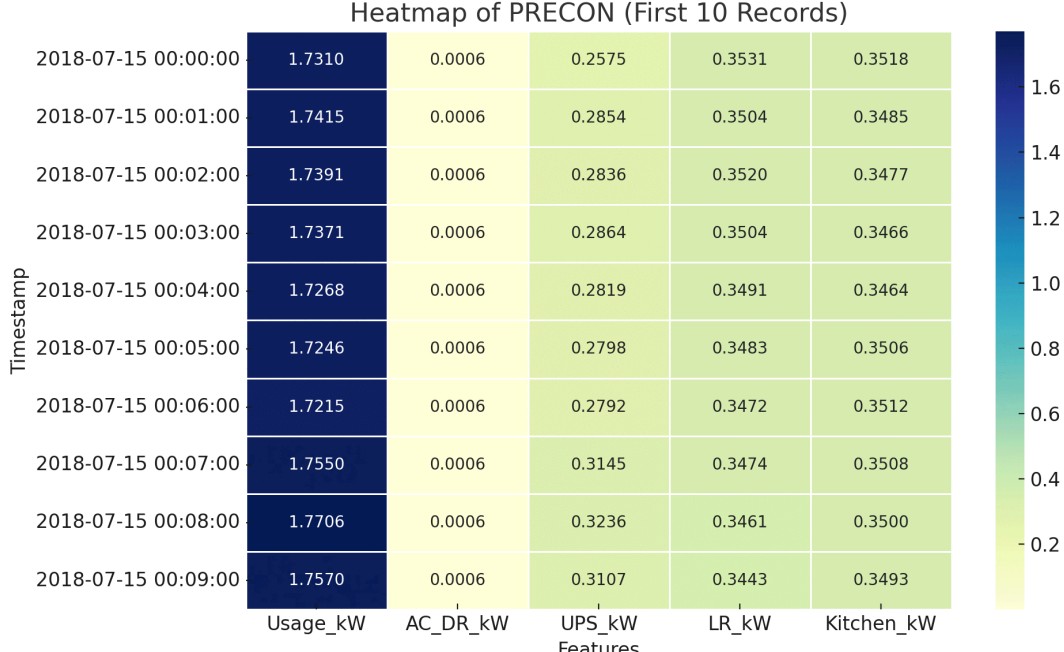

**Fig 7. Heatmap of PRECON (single household).**

for each dataset file. The UUID-based randomization was done using the UUIDv4 library as this ensured the generation of identifiers that are random as well as collision free while also supporting secure indexing, pseudonymity and integrity of client records. These identifiers were then added to all the dataset files with each dataset having a unique ClientID and HouseNumber as shown in Fig 8. The final step is the aggregation of all the clustered dataset files into a single aggregated file. The overall handling of the dataset files as well as the aggregation was done using the Pandas library as it is extremely effective in terms of dataset loading, pre-processing and tabular manipulation.

**Data encryption and blockchain layer.** Once the aggregated dataset from the previous layer is received in the blockchain layer, the ClientID and HouseNumber identifiers are isolated from the dataset and encrypted using salted hash using the SHA256 hash function. The use of salted hash was done using the hashlib library as it enables secure anonymization of the identifiers.

During the hashing of the identifiers, a local blockchain environment is deployed using Ganache, which is basically a local Ethereum test network that provides deterministic accounts, private and public keys and a high speed execution environment. The framework is connected to the local chain using a Remote Procedure Call (RPC) endpoint which in turn could be accessed through a Web3 library. Once the local blockchain was created, two solidity smart contracts (PrivateLayer, ProtectedLayer) were compiled and deployed using the Truffle toolchain. The purpose of each of the smart contracts was to store the salted hashed identifiers.

1. **Private Layer Smart Contract**

    The PrivateLayer smart contract is used to maintain an on-chain registry of the salted HouseNumber. Whenever a new hash is submitted using the PrivateLayer contract, the contract checks whether the HN has already been stored or not. If the HN has not been stored, a new hash is mapped, and an event is emitted. A read only verification function also allows external components to verify whether a particular HouseNumber hash has been recorded previously without the revealing of the original house number. A detailed working of the PrivateLayer smart contract is shown in Algorithm 2.

2. **Protected Layer Smart Contract**

    The ProtectedLayer smart contract is used for the storing of the salted ClientID. Through this contract, each new ClientID hash is validated for uniqueness to prevent duplication or replay. Once a new hashed ClientID is registered successfully, similar to the PrivateLayer, a mapping is recorded, and an event is emitted. A verification function applies a constant time check to verify whether a ClientID hash already exists on-chain or not. This function enables identity traceability while keeping the raw ClientID off-chain. Algorithm 3 provides a descriptive working of the ProtectedLayer smart contract.

| Date_Time | Usage_kW | AC_DR_kW | UPS_kW | LR_kW | Kitchen_kW | AC_Dr_kW | AC_BR_kW | ClientID | HouseNumber |
|---|---|---|---|---|---|---|---|---|---|
| 7/15/2018 0:00 | 1.731 | 0.0006 | 0.2575 | 0.3531 | 0.3518 | 0.0097 | 0.0001 | CID-a2bbc74c | HN-00ed2ed3 |
| 7/15/2018 0:01 | 1.7415 | 0.0006 | 0.2854 | 0.3504 | 0.3485 | 0.0097 | 0.0002 | CID-a2bbc74c | HN-00ed2ed3 |
| 7/15/2018 0:02 | 1.7391 | 0.0006 | 0.2836 | 0.352 | 0.3477 | 0.0097 | 0.0002 | CID-a2bbc74c | HN-00ed2ed3 |
| 7/15/2018 0:03 | 1.7371 | 0.0006 | 0.2864 | 0.3504 | 0.3466 | 0.0096 | 0.0002 | CID-a2bbc74c | HN-00ed2ed3 |
| 7/15/2018 0:04 | 1.7268 | 0.0006 | 0.2819 | 0.3491 | 0.3464 | 0.0097 | 0.0002 | CID-a2bbc74c | HN-00ed2ed3 |
| 7/15/2018 0:05 | 1.7246 | 0.0006 | 0.2798 | 0.3483 | 0.3506 | 0.0098 | 0.0002 | CID-a2bbc74c | HN-00ed2ed3 |
| 7/15/2018 0:06 | 1.7215 | 0.0006 | 0.2792 | 0.3472 | 0.3512 | 0.0098 | 0.0001 | CID-a2bbc74c | HN-00ed2ed3 |
| 7/15/2018 0:07 | 1.755 | 0.0006 | 0.3145 | 0.3474 | 0.3508 | 0.0096 | 0.0003 | CID-a2bbc74c | HN-00ed2ed3 |
| 7/15/2018 0:08 | 1.7706 | 0.0006 | 0.3236 | 0.3461 | 0.35 | 0.0093 | 0.0002 | CID-a2bbc74c | HN-00ed2ed3 |
| 7/15/2018 0:09 | 1.757 | 0.0006 | 0.3107 | 0.3443 | 0.3493 | 0.0095 | 0.0003 | CID-a2bbc74c | HN-00ed2ed3 |

**Fig 8**. **Aggregated dataset with client ID and house number.**

**Algorithm 2 PrivateLayer: House hash registration and verification.**

```
   State: submittedHouseHashes : mapping(string → bool)
2: Event: HouseHashSubmitted(houseHash)
   function SUBMITHOUSENUMBER(houseHash)
4:    Input: houseHash (salted SHA-256 hash of the house identifier)
      Output: None (writes hash to blockchain state)
Require: houseHash ≠ ""
Require: submittedHouseHashes[houseHash] = false
6:    submittedHouseHashes[houseHash] ← true
      emit HouseHashSubmitted(houseHash)
8:    return
   end function
10: function VERIFYHOUSEHASH(houseHash)
      Input: houseHash (salted SHA-256 hash)
12:    Output: exists ∈ {true, false}
      return submittedHouseHashes[houseHash]
14: end function
```

**Algorithm 3 ProtectedLayer: Client hash registration and verification.**

```
1 : State: submittedClientHashes : mapping(string → bool)
2: Event: ClientHashSubmitted(clientHash)
3: function SUBMITCLIENTID(clientHash)
4:    Input: clientHash (salted SHA-256 hash of the client identifier)
5:    Output: None (writes hash to blockchain state)
Require: clientHash ≠ ""
Require: submittedClientHashes[clientHash] = false
6:    submittedClientHashes[clientHash] ← true
7:    emit ClientHashSubmitted(clientHash)
8:    return
9: end function
10: function VERIFYCLIENTHASH(clientHash)
11:    Input: clientHash (salted SHA-256 hash)
12:    Output: exists ∈ {true, false}
13:    return submittedClientHashes[clientHash]
14: end function
```

After the transaction submission of both the ClientID and HouseNumber to their respective smart contracts finishes, the blockchain provides a successful receipt that provides that the transaction has been mined at the end of the blockchain pipeline. Also provided is the gas utilized, execution time as well as latency that would be used for performance analysis later. Once the hashed identifiers are received after passing through the blockchain, these hashed identifiers are placed in the aggregated dataset, replacing the raw identifiers. This results in an encrypted dataset with hashed identifiers and the original energy features.

**Cognitive layer.** The final layer is the cognitive layer which receives the encrypted aggregate datasets at a Federated server. At this point, the dataset is divided into various subsets to cater each of the federated clients, initially set to 5 clients for the simulated environment. Once each client receives a subset of the dataset, the columns of ClientID, HouseNumber and Date_Time is removed from each dataset. The target variable is then set to Usage_kW which is then used for supervised load forecasting tasks. Different ML models, such as Random Forest (RF), XGBoost, HistGradient Boosting (HGB), Multi-layer Perceptron (MLP), Support Vector Regression (SVR), Ridge Regression, Linear Regression (LR), and K-Nearest Neighbor (KNN), are then utilized to train the received dataset. These ML models are loaded using the Scikit-Learn library which is also utilized for the different performance metrics considered for this simulation.

In order to simulate an FL based environment, the Flower framework is utilized which enables communication between multiple FL clients and a central server. The reason for the selection of the Flower framework is because unlike other FL based frameworks, Flower provides simulation of multiple clients, secure aggregation of model updates, customization of strategies (e.g. FedAvg, FedAdam, FedProx etc.), integration with existing PyTorch/Tensorflow models. Once a FL client has trained their respective dataset, the Flower framework ensures that only the weights/parameters from each training are transferred to the central server using the FedAvg algorithm. Once different parameters are received at the central server, they are aggregated together to become a global model. After the aggregation is complete, zero-mean Gaussian noise is injected into the global model using NumPy library. This injection of noise ensures the implementation of the central DP mechanism. This noise injection is done after every communication round. Once the noise has been inserted, the noisy global model is then transmitted back to the FL clients for another communication round.

## Performance evaluation metrics

The predictive accuracy of the proposed framework is evaluated by the following statistical metrics:

- Mean Absolute Error (MAE)

$$\text{MAE} = \frac{1}{n}\sum_{i=1}^{n}|y_i - \hat{y}_i| \tag{12}$$

- Mean Squared Error (MSE)

$$\text{MSE} = \frac{1}{n}\sum_{i=1}^{n}(y_i - \hat{y}_i)^2 \tag{13}$$

- Mean Absolute Percentage Error (MAPE)

$$\text{MAPE} = \frac{100\%}{n}\sum_{i=1}^{n}\left|\frac{y_i - \hat{y}_i}{y_i}\right| \tag{14}$$

- Coefficient of Determination ($R^2$)

$$R^2 = 1 - \frac{\sum_{i=1}^{n}(y_i - \hat{y}_i)^2}{\sum_{i=1}^{n}(y_i - \bar{y})^2}, \quad \text{where} \quad \bar{y} = \frac{1}{n}\sum_{i=1}^{n}y_i \tag{15}$$

## Results

As mentioned above, the experimentation for the proposed BeFL framework was conducted in a layered fashion. The first evaluation conducted was the efficiency evaluation of the blockchain layer which was determined through metrics such as average gas used and the average transaction duration. By assessing the privacy preserving capabilities of the proposed framework, we evaluated a set of privacy specific metrics derived from smart contract interactions. First, raw data leakage was examined by analyzing whether any unencrypted ClientID or HouseNumber values were recorded on-chain. The results confirmed a raw data leakage count of zero, indicating complete anonymization of sensitive fields. Salted hashing was employed with a constant 128-bit entropy, ensuring strong resistance against brute-force and dictionary attacks. Furthermore, only anonymized hash values were stored on-chain, resulting in a blockchain visibility score of 0.33. This reflects minimal metadata exposure and alignment with privacy by design principles.

Operational metrics were also considered. The average gas consumption per transaction was moderate, and the average estimated transaction cost was approximately ~0.000526 ETH for private layer as well as for protected later. Overall, the evaluation confirms that the blockchain layer achieves both privacy preservation and efficient on-chain transaction logging in the proposed architecture.

The gas usage evaluation was conducted for both smart contracts of the proposed framework and was distributed based on average gas usage and average duration as presented in Fig 9 and Fig 10.

As it can be seen in the above Fig 9, the average execution time for each transaction remains consistently low with the private layer showcasing an average of 0.0322 seconds while the protected layers shows an average of 0.0329 seconds. These almost identical times represent that the cryptographic functions are optimized for speed which are extremely important for real time environment scenarios like smart grids.

The above Fig 10 presents that both layers of the proposed framework maintain a constant average gas usage of around 39,000 units. As Ethereum-based systems are sensitive to gas costs, this result shows that the constructed smart contracts are balanced in terms of minimization of overhead while also preserving the robustness of the framework.

The Fig 11 provides a visual evaluation of the mapping of gas usage against latency of both blockchain layers. The result shows that there is negligible correlation which implies that the performance of both layers scale perfectly, even with varied computational loads. The results also present that both layers withstand small spikes in usage without the degradation of speed.

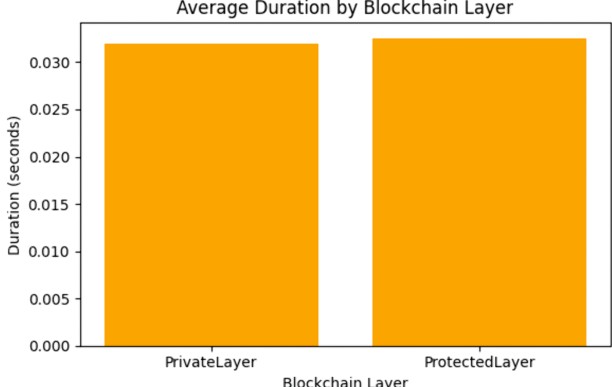

**Fig 9**. **Average execution time by blockchain layers.**

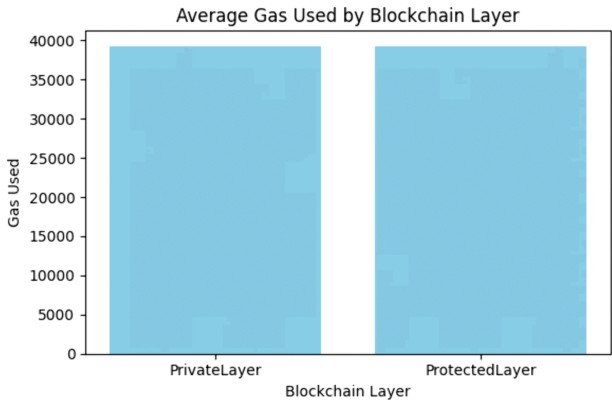

**Fig 10**. **Average gas usage by blockchain layers.**

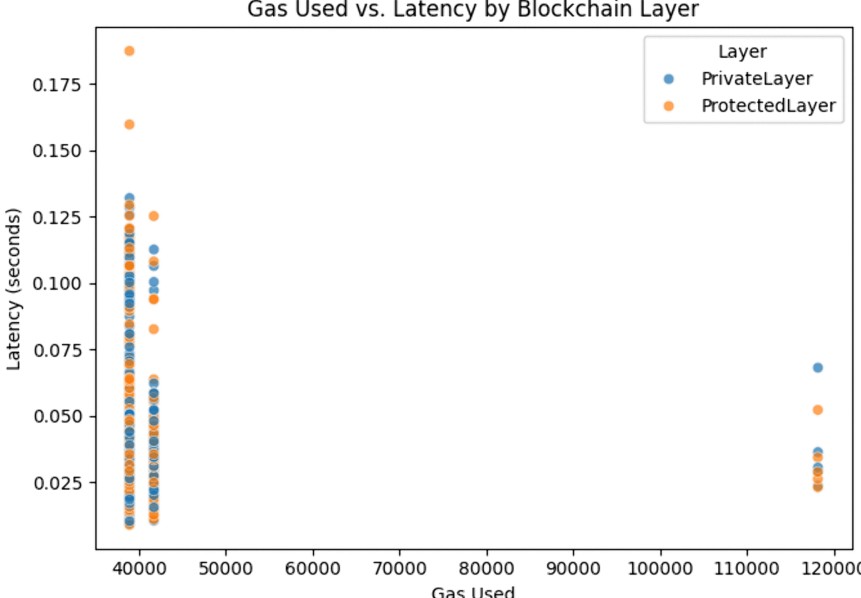

**Fig 11**. **Comparison of gas usage and latency of blockchain layers.**

Fig 12 provides a thorough representation of the latency distribution and frequency of transactions. Fig 12(a) shows a tight inter-quartile range which represents consistent execution times with a few outliers present. While Fig 12(b) is presented as a KDE-augmented histogram which shows that both blockchain layers face a sharp peak around 0.03 seconds which in turn indicates deterministic behavior under normal conditions.

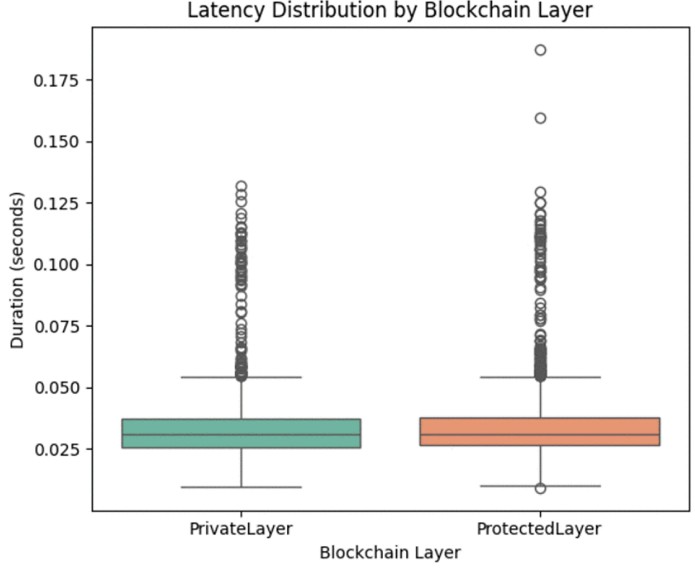

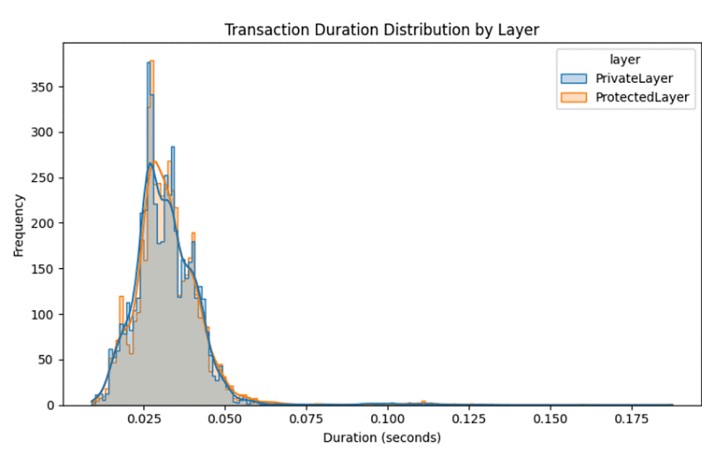

(a) Latency Distribution by Blockchain Layer

(b) Transaction Duration Distribution by Blockchain Layer

**Fig 12**. **Comparison of latency and duration distributions across blockchain layers.**

Considering the above discussed results, it can be seen that the proposed system has a sound ability to hash, store and validate transactions without excessive resource use and without any delay which makes it a good choice for decentralized smart grids. A complete comparison of the different evaluation metrics for both blockchain layers and their implications is presented in Table 4.

The results in Table 4 demonstrate that the dual layer blockchain achieves enhanced privacy and modularity without introducing computational overhead. Both layers show nearly identical execution times (0.0322 vs. 0.0329 s), stable latency (~0.030–0.031 s), and gas efficient performance (~39,100 units). The validity of these results is supported by consistent metrics and the weak correlation between gas usage and latency, confirming stable and reliable performance. Furthermore, because these parameters are platform agnostic, the findings are generalizable to other IoT and cyber-physical systems beyond smart grids. Thus, the dual-layer blockchain provides security and privacy advantages while maintaining efficiency and scalability.

After the evaluation of the proposed blockchain layer, the next phase of evaluation was done in regards to FL process. The resultant aggregated and encrypted dataset was received by an edge node which trained different ML models and then transmitted the model weights to a central server for aggregation, introduction of gaussian noise and the update of the global model. To evaluate the forecasting performance of the proposed framework, we tested multiple regression models on the task of predicting next-hour household energy consumption using the PRECON dataset. The evaluation metrics used include mean squared error (MSE), mean absolute error (MAE), mean absolute percentage error (MAPE) and R-squared ($R^2$) were utilized as shown in Fig 13 measured across three federated learning rounds.

Considering MAE, a lower value would indicate better predictive performance by a ML model. Based on this and shown in Fig 13(a), it can be seen that RF, HGB and XGBoost performed the best, in comparison to the other ML models, which showcases better performance in minimizing absolute deviations between actual and predicted values. On the other hand, the worst performing ML models were Ridge regression and linear regression by demonstrating higher MAE values which indicated poor estimation accuracy.

Fig 13(b) provides a representation of MAPE, where a lower value would present better relative error performance. Through this evaluation, the best performing ML models were again RF, XGBoost and HGB, all of which outperformed the other ML models, having minimal percentage errors. In contrast, ridge regression and MLP performed the worst showing higher MAPE values which indicated suboptimal performance under percentage based evaluations.

In terms of MSE, as shown in Fig 13(c), a lower value is desirable as MSE penalizes larger errors more significantly. Based on this criteria, the best performing ML models were RF and XGBoost which performed with the lowest MSE values, followed closely by gradient boosting. The worst performing ML models were ridge regression and SVR which yielded the highest MSE which suggested that the predictions of these models were associated with higher variance form the true values. Tree-based models such as RF and XGBoost demonstrated superior performance compared to linear models like LR and Ridge Regression. This performance difference is attributed to the non-linear and hierarchical nature of energy consumption data, which often reflects complex user behavior, time-based usage patterns, and environmental

**Table 4**. Blockchain layers evaluation metrics comparison.

| Evaluation Metric | Private Layer | Protected Layer | Insights |
|---|---|---|---|
| Average Duration(s) | 0.0322 | 0.0329 | Almost identical execution across both layers |
| Average Gas Used | 39151 | 39137 | Gas-efficient smart contracts |
| Latency Distribution (Median) | ~0.030 sec | ~0.031 sec | Stable distribution with minimal outliers |
| Outliers in Latency | Few | Few | Represents consistent and robust performance |
| Transaction Duration Speed | 0.02 − 0.05 sec | 0.02 − 0.05 sec | Predictive transaction behavior |
| Correlation of Gas vs Latency | Weak | Weak | Variation in gas does not affect transaction time |
| Security Mechanism | SHA-256 (House Numbers) | SHA-256 (Client IDs) | Hashing in both layers to ensure privacy |
| Role | Identity Masking | Access Control | Enhancement of modularity due to functional isolation |

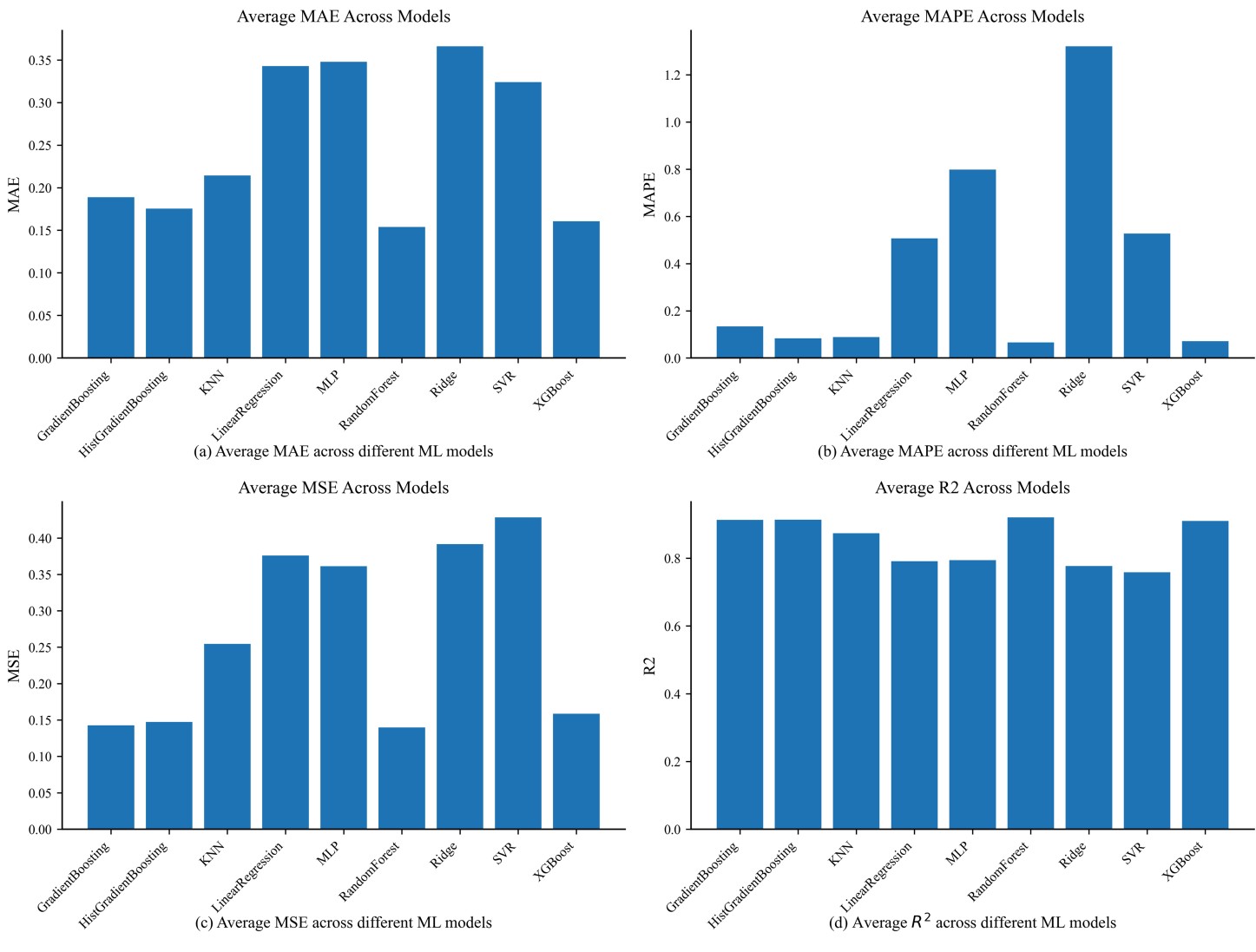

**Fig 13**. **Comparison of ML models using different evaluation metrics.**

factors. Unlike linear models, tree-based methods can automatically detect interactions between features, adapt to heterogeneous input distributions, and handle non-linear dependencies more effectively. These characteristics make them particularly well-suited for modeling time-series energy data across diverse households.

As shown in Fig 14, ensemble tree-based models such as RF and XGBoost consistently outperformed linear and kernel-based models across all metrics. RF achieved the lowest MAE ($\sim$0.153), lowest MSE ($\sim$0.143), and highest $R^2$ score ($\sim$0.92), closely followed by XGBoost with comparable results. These models are better suited for this task as they effectively capture non-linear dependencies and feature interactions present in time-series energy data. Linear models (e.g., LR, Ridge, SVR) showed higher MAE and MAPE values, indicating limited ability to model temporal or contextual variations in the data. For instance, Ridge regression resulted in the highest MAPE ($>$1.3) and one of the lowest $R^2$ scores ($\sim$0.78), confirming its inability to adapt to non-linearity in consumption behavior. Neural models such as MLP underperformed compared to tree-based models, potentially due to limited training data per client in federated rounds, which

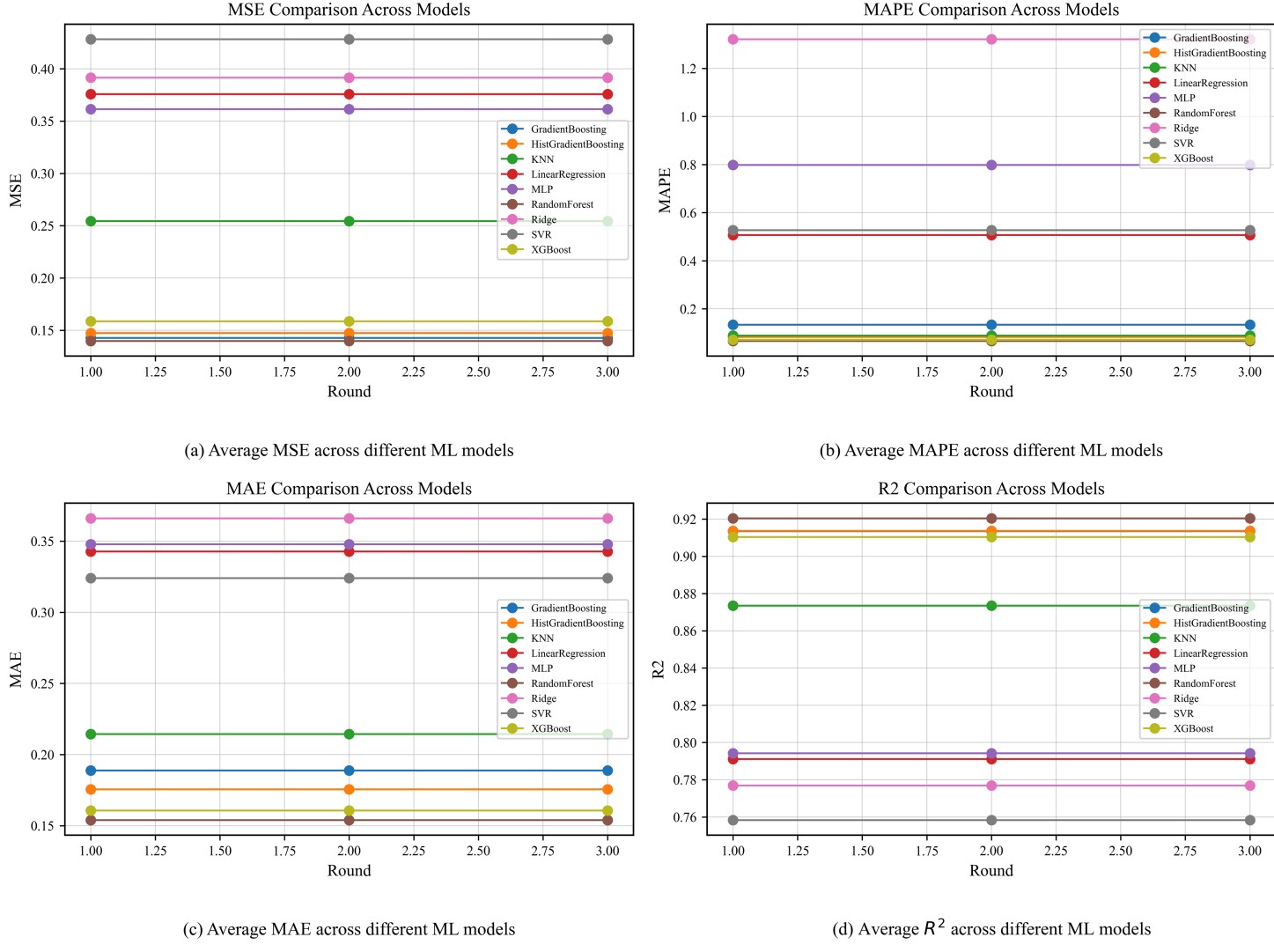

(a) Average MSE across different ML models

(b) Average MAPE across different ML models

(c) Average MAE across different ML models

(d) Average $R^2$ across different ML models

**Fig 14**. **Comparison of ML models using different evaluation metrics.**

affects convergence. Additionally, KNN delivered moderate performance but showed relatively higher error variance, making it less suitable for time-sensitive load forecasting. Across all five rounds, the metrics remained stable, indicating consistent convergence and robustness of the federated training process. The low variation also confirms that the injected noise from the central differential privacy mechanism did not significantly degrade model performance.

The results of these model comparison are compiled and represented in Table 5. This table presents the predictive performance of various machine learning models evaluated on the next-hour energy load forecasting task. Lower MAE, MAPE, and MSE values indicate better performance, while higher $R^2$ values indicate stronger model fit. Table 5 shows that our innovation preserves high forecasting accuracy while adding privacy and security. Tree-based models (Random Forest, XGBoost) achieved the best results ($R^2 > 0.91$), confirming that the framework does not compromise model utility. The validity of the findings is supported by consistent performance across metrics and alignment with the non-linear

**Table 5**. Comparison of model performance for next-hour energy load forecasting.

| Model | MAE | MAPE | MSE | $R^2$ |
|---|---|---|---|---|
| Random Forest | 0.153 | 0.085 | 0.143 | 0.920 |
| XGBoost | 0.158 | 0.092 | 0.157 | 0.914 |
| HistGradientBoosting | 0.172 | 0.101 | 0.151 | 0.911 |
| Gradient Boosting | 0.186 | 0.137 | 0.145 | 0.912 |
| K-Nearest Neighbors (KNN) | 0.216 | 0.104 | 0.255 | 0.874 |
| Multi-Layer Perceptron (MLP) | 0.345 | 0.800 | 0.363 | 0.792 |
| Linear Regression | 0.343 | 0.510 | 0.378 | 0.791 |
| Support Vector Regression (SVR) | 0.329 | 0.540 | 0.435 | 0.759 |
| Ridge Regression | 0.364 | 1.310 | 0.392 | 0.778 |

nature of energy data. Since standard regression metrics were used on realistic smart grid data, the results are generalizable to other energy forecasting and IoT contexts. Thus, the framework delivers secure and privacy-preserving forecasting without sacrificing accuracy.

## Comparison of proposed framework with current literature

In order to make the contribution of the proposed framework contextual, the comparison was done relative to four recent works incorporating blockchain and FL in smart grid and IoT applications. The works that are chosen are Zhou et al. [9], Alam et al. [10], Otoum et al. [11] and Antal et al. [38]. All these studies have a different look into the privacy and security maintenance in a distributed setting, but significant differences also occur in their level, mechanics, and effectiveness. Antal et al. [38] presented a smart grid forecasting framework that uses blockchain to guarantee on-chain replication of the global model thus enabling immutability and provenance guarantees. This makes auditability stronger but also makes model updates vulnerable to inference attacks. Zhou et al. [9] introduced a blockchain assisted data aggregation protocol that uses Shamir secret sharing, random masking, as well as local DP at the query level with robust privacy in the presence of meter and server failures, but without assuring any privacy protections beyond aggregation. Alam et al. [10] presented the conceptual Blockchain-Federated Reinforcement Learning (B-FRL) model, in which the transactions of model-update are secured with blockchain, yet they did not present a specific design of differential privacy and secure aggregation. Otoum et al. [11], lastly, proposed a FL framework facilitated by blockchain in critical IoT, which utilized reinforcement learning to perform client selection and trust scoring, and also used blockchain miners to validate local model updates. Despite being successful at improving both the trustworthiness of devices and the network lifetime, the framework does not provide the framework of privacy hardening the update level. On the other hand, the proposed framework offers a defense-in-depth construction that assuages the weaknesses of such approaches. Secondly, it does not entirely replicate the model on-chain since it stores only hash identifiers and cryptographic proofs providing less attack surface than [11,38]. It also opposed to [9], where DP is applied at a query stage only, the proposed system adds noise to DP during the client-update stage, preventing membership inferences and model inversions attack prior to aggregation. While [10] highlights blockchain's role in securing transactions, the proposed BeFL framework extends this by employing a two-layer smart contract mechanism (PrivateLayer and ProtectedLayer) to separately manage hashed Client IDs and House Numbers, preventing single-contract re-identification. Moreover, secure aggregation and granular access control ensure resilience against collusion and malicious aggregators, while support for dynamic participation without re-keying enables adaptability under device churn. These features collectively demonstrate that the proposed framework surpasses existing works in terms of privacy preservation, security assurance, and operational robustness. The proposed framework is strengthened because it follows a multi-layered blockchain in combination with FL and central DP for secure aggregation. In addition, using DP at the client-update level, the framework can actively overcome the membership inference and model inversion

risks, which are not covered exhaustively in [9,10]. A key novelty is the introduction of a two-layer smart contract mechanism (PrivateLayer and ProtectedLayer), which separates hashed Client IDs and House Numbers across distinct contracts, effectively preventing single-contract breaches from compromising identity privacy. Thirdly, unlike [9], which applies DP only at the query stage, the proposed system implements DP noise addition at the client-update level, mitigating membership inference and model inversion attacks before aggregation. These characteristics positioned the proposed framework as a more privacy-efficient, secure, and resilient smart grid and critical IoT infrastructures solution than current methods in the context of privacy. Table 6 summarize the comparative analysis of security and privacy features across related works and the proposed framework .

## Discussion

In this study, the proposed work focuses on the design of a multi-layer blockchain–enabled federated learning framework with central differential privacy for smart grid energy forecasting, addressing security and privacy together rather than in isolation. The evaluation criteria were defined across two dimensions:

- security and privacy metrics (execution time, gas usage, latency, hashing, and identity protection) and
- forecasting performance metrics (MAE, MAPE, MSE, $R^2$).

The evaluation process included deploying Ethereum-based smart contracts to assess blockchain performance, training multiple machine learning models under the proposed BeFL setup, and benchmarking against related works. The results demonstrated that the dual-layer blockchain introduced enhanced confidentiality and modularity without overhead (~0.032s execution, ~39k gas) and that tree-based models maintained high forecasting accuracy ($R^2 > 0.91$), showing that the framework preserves utility while adding strong security. From the gap analysis, it is evident that prior works relied on single-chain storage, limited DP, or failed to jointly evaluate privacy and model accuracy, whereas our approach integrates dual-layer blockchain, central DP, and FL to overcome these limitations. However, the current approach combines dual-layer blockchain, central differential privacy and federated learning, which overcomes these shortcomings. Consequently, we recommend the application of a dual layer blockchain architecture with the central differential privacy for privacy sensitive and decentralized environments because of its scalability and the ability to fit a wider range of IoT based applications that go beyond advanced smart grids.

Table 6. **Comparative analysis of key security and privacy features across related works and the proposed framework.**

| Feature / Research Papers | [38] | [9] | [10] | [11] | **Proposed Framework** |
|---|---|---|---|---|---|
| **Blockchain-based Storage** | Basic Blockchain | Hyperledger Fabric | Blockchain for IoT | Blockchain for FL | Multi-layer blockchain with private & protected layers |
| **Differential Privacy (DP)** | X | Local DP for aggregation results | X | X | DP integrated at model & result level |
| **End-to-End Confidentiality** | Encryption | Random masking + Shamir | Encryption in IoT data | Secure model updates | Multi-layer encryption + DP + Blockchain |
| **Aggregation Scheme** | Homomorphic Encryption | Shamir Secret Sharing + Random Masking | Symmetric Encryption | FL model aggregation | FL aggregation with DP and Blockchain |
| **Anonymity / Identity Protection** | ~ Basic pseudonyms | Identity masking | X | X | Identity hashing + blockchain ledger separation |

## Impact of data heterogeneity and bias

A significant feature of real-world smart grid scenarios is that there is data heterogeneity, i.e. each household or smart meter can have distinct consumption patterns, usage behaviors and machine-level variations. Although the existing implementation of the framework uses the IID partitioning of the PRECON dataset to set a baseline performance and test the core functionality, the federated learning design would naturally provide decentralized learning using heterogeneous data. The use of the Federated Averaging (FedAvg) algorithm mitigates the influence of local biases by proportionally weighting client updates based on dataset size, promoting generalizable learning over overfitting to localized anomalies. In addition, the incorporation of central differential privacy during the aggregation phase introduces Gaussian noise, which statistically smooths extreme or skewed updates, offering additional robustness against client-specific biases. Moreover, the implementation of central differential privacy at the aggregation step also brings statistical smoothing of Gaussian noise that weakens the effect of extreme updates or skewed updates of individual clients. Even though the heterogeneity of the data was not directly represented with the help of the model in this paper, the next steps will be to simulate non-IID data and add personalization strategies or clients selection mechanisms to the framework in order to ensure the enhancement of the resilience to the variability of the real smart grid.

## Robustness and explainability

The proposed framework strengthens both the robustness and explainability of FL by integrating multi-layer safeguards with auditable transparency mechanisms. Robustness is achieved by norm clipping, which basically bounds the magnitude of each FL client local model updates and prevents any client, either faulty or malicious, from inordinately influencing the global model, thus mitigating poisoning risks. Also, client participation is restricted to only those whose identities have been securely registered via the BeFL framework blockchain, which not only ensures authentication but also reduces the likelihood of unauthorized model submission and Sybil attacks. Furthermore, during the aggregation phase of local updates, through the employed FedAvg algorithm strategy, the updates are weighed according to the relative size of each client dataset which potentially filters out inconsistent deviant updates. These approaches ensure that the central server is able to mitigate the influence of nodes that are compromised and guarantee that the training process remains reliable.

The aspects of explainability and transparency is achieved by the BeFL framework in the blockchain phase where the ClientID and House Number identifiers are first salted and then hashed. These salted and hashed identifiers are then transferred into the blockchain. Through the maintenance of each round records involving model updates, client participation, and details of aggregations, stakeholders are able to perform post-hoc analysis. From this analysis, it is possible to monitor performance fluctuations and anomalies of specific rounds or involved clients. From these methods, it is possible to ensure that the process of model training remains transparent for regulatory compliance and accountability.

## Critical trade-offs

The proposed BeFL framework has aimed to create a balance between maintaining client privacy and preservation of model accuracy by incorporating central DP. This is done by injecting Gaussian noise to the aggregated model at the cognitive layer in order to hide the identification of all the participating clients. Through this approach, the risk of data leakage is mitigated while also reducing the risk of reconstruction attacks. However, with the injection of noise, a loss of model performance was observed. In order to counter this issue, the BeFL framework utilizes the possibility of adjustment of noise levels according to the sensitivity of the dataset distribution and aggregation function. Another approach utilized was the clipping of local model updates to a predefined norm which limited the variance before the injection of Gaussian noise. These approaches ensured that the BeFL performed relatively better as seen in MSE and MAE, discussed in previous section, while still preserving the privacy of clients and adhering to regulatory compliance.

## Real world implications and applications

The proposed framework BeFL includes federated learning, identity protection with the help of blockchain, and differential privacy. Other than SG, it has significant applicability in various areas where sensitive distributed data is utilized. In healthcare, for instance, hospitals or diagnostic centers could collaboratively train predictive models on patient data without sharing raw medical records, while blockchain ensures traceable participation and privacy preserving auditability. In finance, local records of transactions could be used to identify fraud or evaluate risk through the decentralized banking or insurance providers who protect the identities of clients. Likewise, in industrial IoT contexts, where confidentiality and integrity of data are the key concerns, the framework can support secure cooperation of distributed edge devices. The layered architecture has a modularity that allows adopting domain specific needs and makes it a flexible system to solve the privacy preserving, trustful, and decentralized machine learning in controlled industries.

## Limitations and future work

Although the proposed federated learning framework with blockchain integration provides significant capabilities with respect to preserving privacy and providing exact analytics with respect to energy usage, several limitations became apparent in the course of implementation. Experimental validation was done as IID conditions, but realistic smart grid data mostly contain non-IID characteristics with respect to variable consuming patterns. It may have adverse effect on the federated model with regard to convergence and generalization of the model. The blockchain layer, although providing secure and unalterable logging, adds latency and increases gas costs. While these metrics stayed within acceptable limits during simulations, scaling up to larger implementations could intensify these issues. Adjusting differential privacy parameters indicated a trade-off between privacy preservation and model accuracy which showed a need for future research into adaptive privacy schemes. Moreover, anomalous residuals seen for specific clients even more strongly illustrated the influence of uneven data distributions that indicated the need for more advanced client-side training adaptations.

Future work should investigate methodologies to provide better support for non-IID data to increase the scalability and resilience in heterogeneous data distributions. This includes adopting resource aware model training on edge devices and looking into efficient blockchain alternatives. Additionally, the integration of advanced privacy preservation techniques, such as homomorphic encryption or secure aggregation, as well as the extension of the framework to handle deep learning models for time series forecasting is something that will certainly increase its applicability to practical smart grid scenarios.

## Conclusion

The paper presented a novel privacy preserving framework utilizing smart grid combining federated learning with This paper has presented a privacy preserving framework of smart grids with the combination of federated learning, central differential privacy and a multi-layer blockchain. The framework prevents third parties access to consumer information by keeping raw processing local to the client and storing cryptographical hashes on the blockchain, thus allowing the confidentiality of the processed data as well as verifiability. The introduction of an adaptive DP-FedAvg is further beneficial in terms of enhanced privacy guarantees and preservation of similar predictive performance. Experimental evaluations show that ensemble models like Random Forest and XGBoost get a higher forecasting accuracy than linear models because they can capture the non-linearity of energy consumption data. The reduction in the accuracy due to the introduction of the DP noise is relatively low, making this trade-off very acceptable and essential to secure consumers data. Through this balance, it is possible to see how efficient the proposed framework BeFL can be to solve this long standing dilemma of balancing privacy and feasible load forecasting. In summary, the proposed approach not only safeguards sensitive consumer data but also offers a scalable and secure architecture suitable for real world deployment in smart grids. Future work will explore adaptive noise mechanisms, integration with edge computing optimizations, and evaluation on larger, heterogeneous datasets to further enhance both robustness and applicability.

## Author contributions

**Conceptualization:** Fatima Tariq.

**Data curation:** Fatima Tariq.

**Formal analysis:** Fatima Tariq.

**Funding acquisition:** Xiaochun Cheng.

**Investigation:** Fatima Anjum, Nadia Kanwal.

**Methodology:** Nadia Kanwal.

**Resources:** Fatima Anjum.

**Supervision:** Fatima Anjum, Nadia Kanwal.

**Validation:** Xiaochun Cheng, Shazia Javed.

**Visualization:** Khursheed Aurangzeb.

**Writing – original draft:** Fatima Tariq.

**Writing – review & editing:** Fatima Anjum, Xiaochun Cheng, Shazia Javed, Khursheed Aurangzeb, Nadia Kanwal.

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
