## [Decision Letter · Decision Letter 0]

6 Nov 2025

PONE-D-25-52221Towards a Cybersecure and Privacy Enhanced Smart Grid: A Blockchain Enabled Federated Learning FrameworkPLOS ONE

Dear Dr. Cheng,

Thank you for submitting your manuscript to PLOS ONE. After careful consideration, we feel that it has merit but does not fully meet PLOS ONE’s publication criteria as it currently stands. Therefore, we invite you to submit a revised version of the manuscript that addresses the points raised during the review process.

We look forward to receiving your revised manuscript.

Kind regards,

Dr Hasan Tahir

Academic Editor

PLOS ONE

**Journal Requirements:**

“The authors have been funded by UKRI EPSRC  Grant EP/W020408/1 Project SPRITE+ 2: The Security, Privacy, Identity and Trust Engagement Network plus (phase 2) for this study.

The authors also have been funded by PhD project RS718 on Explainable AI through UKRI  EPSRC Grant funded Doctoral Training Centre at Swansea University.”

“This work was supported by UKRI EPSRC Grant funded Doctoral Training Centre at 740 Swansea University, through PhD project RS718 on Explainable AI. 741 Authors also have been supported by UKRI EPSRC Grant EP/W020408/1 Project 742 SPRITE+ 2: The Security, Privacy, Identity and Trust Engagement Network plus 743 (phase 2).”

“The authors have been funded by UKRI EPSRC  Grant EP/W020408/1 Project SPRITE+ 2: The Security, Privacy, Identity and Trust Engagement Network plus (phase 2) for this study.

6. We note that Figures 1, 2, 3, 4, and 5 in your submission contain copyrighted images. All PLOS content is published under the Creative Commons Attribution License (CC BY 4.0), which means that the manuscript, images, and Supporting Information files will be freely available online, and any third party is permitted to access, download, copy, distribute, and use these materials in any way, even commercially, with proper attribution. For more information, see our copyright guidelines: http://journals.plos.org/plosone/s/licenses-and-copyright.

a. You may seek permission from the original copyright holder of Figure(s) [#] to publish the content specifically under the CC BY 4.0 license.

**Additional Editor Comments:**

I have gone through the detailed comments provided by the worthy reviewers. The research paper needs further refinement and extensions as currently there are gaps in the presentation. Please revise the paper in light of the reviewer comments addressing the concerns as required. Please focus on readability and the overall flow of the presentation while incorporating the necessary changes.

**Reviewers' comments:**

Reviewer's Responses to Questions

**Comments to the Author**

1. Is the manuscript technically sound, and do the data support the conclusions?

Reviewer #1: Yes

Reviewer #2: Yes

2. Has the statistical analysis been performed appropriately and rigorously?

Reviewer #1: Yes

Reviewer #2: Yes

3. Have the authors made all data underlying the findings in their manuscript fully available?

Reviewer #1: Yes

Reviewer #2: Yes

4. Is the manuscript presented in an intelligible fashion and written in standard English?

Reviewer #1: Yes

Reviewer #2: Yes

5. Review Comments to the Author

Reviewer #1: 1) This paper proposes a secure and privacy-preserving framework that integrates a dual layer blockchain architecture, federated learning (FL) and central differential privacy (DP) to address challenges related to user privacy and system-level security holistically.

-- In the abstract, you need to mention the research gaps in the current techniques for addressing these user privacy and system-level security challenges. This should be done prior to introducing the proposed framework.

2) To mitigate against reader confusion, all acronyms must be written in full the first time they appear within text. For instance, you have used ' MAE, MAPE, MSE' in the abstract devoid of stating what they stand for in full.

3) What is the need for 'Author summary' section?

4) The Introduction is not elaborate enough so as to bring out the research area and problem domain. Therefore, you need to expand this section so that you adequately describe the research area and problem domain.

5) The 'Problem Statement' section is not well done as you have began by stating that 'These vulnerabilities can be addressed by incorporating a blockchain-based security layer that ensures tamper-proof storage, immutable audit trails, and secure identity protection through the use of salted cryptographic hashes.'

- It is not clear which vulnerabilities you are alluding to.

- This section is not written in a logical and concise manner.

6) The 'Key Contributions ' is not well done. Th main/key contributions must be stated in point form; with each of these points reflecting some kind of notable novelty.

7) In the 'Literature Work' section, you have failed to bring out the research gaps in each of the presented works. It is therefore extremely difficult to discern which gaps this study sought to bridge.

8) Consider expanding Table 1 to include the limitations of the reviewed works.

9) Immediately after Table 1, add a paragraph to summarize the identified research gaps. Thereafter, explain how the developed framework helps bridge these gaps.

10) To enhance readability, add a table of all notations/symbols used throughout this paper.

11) Algorithm 1 should be well described within text. In addition, it should have well defined input (s) and output (s). The same applies to all figures within this paper.

12) In the Experimental Setup section, give rationale for the selected tools/values/settings/libraries

13) Implementation details of the Federated Learning have not been adequately described. The same can be said of Data Pre-processing.

14) Give the mathematical formulations of the deployed metrics, such as MAPE,MAE & MSE.

15) The clarity of all figures must be improved.

16) Describe how the identified limitations of the proposed framework can be addressed.

Reviewer #2: 1. Can you provide more details on the dual-layer blockchain architecture and its implementation?

2. How does the framework handle data heterogeneity and variability in smart grid environments?

3. Can you discuss the trade-off between privacy and accuracy in the proposed framework?

4. How does the framework ensure regulatory compliance with standards such as GDPR and CCPA?

5. Can you provide more information on the Flower framework and its role in implementing the proposed system?

6. How does the framework handle model updates and aggregation in the presence of faulty or malicious clients?

7. Can you discuss the potential applications of the proposed framework beyond smart grids?

8. How does the framework ensure transparency and explainability in FL model updates?

9. Can you provide more details on the Gaussian noise application and its impact on model performance?

10. Highlighted article might be considered for related work. (Dhasarathan, C., Ramachandra, R. B., & Tripathi, D. Safeguarding User Privacy at the Edge of Fog Computing Networks in Decentralized Distributed Computing. In Swarm Intelligence (pp. 133-155). CRC Press.)

11. How does the framework plan to address potential biases and disparities in FL model updates and predictions?

6. PLOS authors have the option to publish the peer review history of their article (what does this mean?). If published, this will include your full peer review and any attached files.

Reviewer #1: No

Reviewer #2: No

---

## [Author Response · Author response to Decision Letter 1]

12 Dec 2025

We have attached Response to Reviewers.pdf

---

## [Decision Letter · Decision Letter 1]

26 Jan 2026

Towards a Cybersecure and Privacy Enhanced Smart Grid: A Blockchain Enabled Federated Learning Framework

PONE-D-25-52221R1

Dear Dr. Cheng,

We’re pleased to inform you that your manuscript has been judged scientifically suitable for publication and will be formally accepted for publication once it meets all outstanding technical requirements.

Kind regards,

Hasan Tahir

Academic Editor

PLOS One

Additional Editor Comments (optional):

After review and no further comments, the paper is recommended for publication.

Reviewers' comments:

Reviewer's Responses to Questions

**Comments to the Author**

1. If the authors have adequately addressed your comments raised in a previous round of review and you feel that this manuscript is now acceptable for publication, you may indicate that here to bypass the “Comments to the Author” section, enter your conflict of interest statement in the “Confidential to Editor” section, and submit your "Accept" recommendation.

Reviewer #1: All comments have been addressed

Reviewer #2: All comments have been addressed

2. Is the manuscript technically sound, and do the data support the conclusions?

Reviewer #1: Yes

Reviewer #2: Yes

3. Has the statistical analysis been performed appropriately and rigorously?

Reviewer #1: N/A

Reviewer #2: Yes

4. Have the authors made all data underlying the findings in their manuscript fully available?

Reviewer #1: Yes

Reviewer #2: Yes

5. Is the manuscript presented in an intelligible fashion and written in standard English?

Reviewer #1: Yes

Reviewer #2: Yes

6. Review Comments to the Author

Reviewer #1: Thank you for adequately responding to the previous comments. This paper is now in great shape and therefore be accepted for publication.

Reviewer #2: The revised article have incorporated all suggested review comments with appropriate scientific answers that suits the proposed system.

7. PLOS authors have the option to publish the peer review history of their article (what does this mean?). If published, this will include your full peer review and any attached files.

Reviewer #1: No

Reviewer #2: No

---

## [Editor Report · Acceptance letter]

PONE-D-25-52221R1

PLOS One

Dear Dr. Cheng,

I'm pleased to inform you that your manuscript has been deemed suitable for publication in PLOS One. Congratulations! Your manuscript is now being handed over to our production team.

Kind regards,

on behalf of

Dr. Hasan Tahir

Academic Editor

PLOS One